



# Quantitative reconstruction of past monsoon precipitation based on tetraether membrane lipids in Chinese loess

Jingjing Guo[1], Martin Ziegler[1], Louise Fuchs[1], Youbin Sun[2], Francien Peterse[1]

[1]Department of Earth Sciences, Utrecht University, 3584 CB Utrecht, the Netherlands
[2]State key laboratory of Loess and Quaternary Geology, Institute of Earth Environment, Chinese Academy of Sciences, Xi'an, 710061, China

*Correspondence to*: Jingjing Guo (j.guo@uu.nl)

**Abstract.** Variations in the oxygen isotope composition ($\delta^{18}O$) of cave speleothems and numerous proxy records from loess-paleosol sequences have revealed past variations in East Asian monsoon (EAM) intensity. However, challenges persist in
reconstructing precipitation changes quantitatively. Here, we use the positive relationship between the degree of cyclization (DC) of branched glycerol dialkyl glycerol tetraethers (brGDGTs) in modern surface soils from the Chinese loess Plateau (CLP) and mean annual precipitation (MAP) to quantify past monsoon precipitation changes on the CLP. We present a new ~130,000-year long DC-based MAP record for the Yuanbao section on the western edge of CLP, which closely tracks the orbital- and millennial-scale variations in both the speleothem $\delta^{18}O$ record and the hydrogen isotope composition of plant
waxes ($\delta^{2}H_{wax}$) from the same section. Combing our new data with existing brGDGT records from other CLP sites reveals a spatial gradient in MAP that is most pronounced during glacials, when the western CLP experiences more arid conditions and receives up to ~250 mm less precipitation than in the southeast, whereas MAP is ~850 mm across the CLP during the Holocene optimum. Furthermore, the DC records show that precipitation amount on the CLP varies at the precession as well as obliquity scale, as opposed to the primarily precession scale variations in speleothem $\delta^{18}O$ and $\delta^{2}H_{wax}$ at Yuanbao, and the
100-kyr cycle in other loess proxies such as magnetic susceptibility, which rather indicates the relative intensity of the EAM. At the precession scale, the DC record is in phase with $\delta^{2}H_{wax}$ from same section as well as the speleothem $\delta^{18}O$ record, which supports the hypothesis that monsoon precipitation is driven by northern hemisphere summer insolation.

## 1 Introduction

The East Asian Monsoon (EAM) is one of the strongest monsoon systems on Earth, and driven by land-sea temperature
contrasts. It consists of a warm and wet summer monsoon (EASM) transporting moisture from low-latitude oceans to the East Asian continent, and a cold and dry winter monsoon (EAWM) that transfers high-latitude cold and dry air to East Asia (Liu and Ding, 1998). Ongoing global warming will likely affect the strength of the EAM and, therefore, the amount of moisture delivered to the continent by the EASM, where it regulates the water supply to over 20% of the world's population (An et al., 2015). Past variations in EAM climate have been inferred based on the oxygen isotope composition ($\delta^{18}O$) of cave
speleothems, which show a dominant precession signal (Wang et al., 2008; Cheng et al., 2016; Wang et al., 2001) and



support the hypothesis that monsoons are primarily driven by northern hemisphere summer insolation (NHSI) (Kutzbach, 1981). However, the 6–8 thousand years (kyr) lag of a stacked summer monsoon record from Arabian Sea to NHSI led to the suggestion that monsoon variations are rather impacted by latent heat influxes from southern hemisphere and global ice volume (Clemens and Prell, 2003). The subsequent observation that speleothem $\delta^{18}O$ also lags NHSI at the precession cycle

was explained by an impact of winter temperature on speleothem $\delta^{18}O$, which should thus not be interpreted as EASM precipitation alone (Clemens et al., 2010). The dominant eccentricity and obliquity cycles that appear in a $\delta^{18}O$ record of planktonic foraminifera offshore the Yangtze River Valley reflecting EAM runoff after removing confounding factors on $\delta^{18}O$, such as temperature, was further presented as evidence that monsoon precipitation is more sensitive to ice volume and greenhouse gas instead of directly reacting to NHSI (Clemens et al., 2018). In addition, the 100-kyr cycles are dominant in

records derived from loess-paleosol sequences from the Chinese Loess Plateau (CLP) in north-central China (Liu and Ding, 1998), where proxies such as magnetic susceptibility (MagSus) (Maher and Thompson, 1991; An et al., 1991), grain size (GS) (Xiao et al., 1995), and carbonate content (Sun et al., 2010) have recorded past variations in EAM climate (Sun et al., 2015). In short, glacials are characterized by a strong EAWM, creating overall cool and dry conditions with loess deposition, reflected by coarser GS and lower MagSus value, whereas interglacials are characterized by a more intense EASM with a

warm and wet climate resulting in intensified soil formation, finer GS and higher MagSus value (Liu and Ding, 1998). Despite the presence of this strong glacial-interglacial cyclicity across the CLP, several newer proxies indicate that the 23-kyr cycle is also recorded in the loess, pointing at NHSI as important driver of the EASM after all (e.g., Fuchs et al., 2023; Guo et al., 2022; Li et al., 2024; Beck et al., 2018). Regardless of this so-called 'Chinese 100-kyr problem', these reconstructions provided mostly qualitative changes in the strength of summer and winter monsoons, and often represent a

mixed signal of both temperature and precipitation (Cheng et al., 2022).

Multiple studies have attempted to quantitatively reconstruct EAM precipitation on the CLP using various proxies. These include for example the Sr/Ca ratio of microcodium, which is based on the principle that the elements incorporated into the structure of micrododium are impacted by the composition changes in soil solution between evolved soil solutes with higher Sr/Ca ratio and fresh water with lower Sr/Ca ratio (Li and Li, 2014). Additionally, phytolith assemblages reflect changes in

vegetation under different moisture conditions (Lu et al., 2007), as does the carbon isotopic composition of total organic carbon ($\delta^{13}C_{TOC}$), which reflects changes in $C_3$ and $C_4$ plants (Vidic and Montañez, 2004; Rao et al., 2013). Meteoric Beryllium ($^{10}Be$) stored in loess-paleosol sequences across the CLP provides another rainfall indicator as it attaches to atmospheric dust particles that are mainly deposited during rainfall events (Beck et al., 2018; Zhou et al., 2023). However, application of these proxies across the CLP does not generate consistent results. For example, precipitation estimates based

on $^{10}Be$ and phytolith assemblages for the southern CLP differ by up to 400 mm during glacials and 200 mm during interglacials (Lu et al., 2007; Beck et al., 2018). Similarly, comparing different proxy estimates of maximum precipitation during interglacials results in a spatial variation of > 800 mm across the CLP (e.g., ~300 mm on the central CLP based on microcodium Sr/Ca (Li et al., 2017) vs ~1200 mm on the southern CLP based on $^{10}Be$ (Beck et al., 2018) during MIS5), far exceeding the modern gradient in mean annual precipitation (MAP) (~500 mm) (Xu and Huang, 2023), whereas the inland



reach of the EASM presumably exceeded the location of the CLP during these intervals. Together with the different orbital

cyclicities represented in these proxy records, this suggests that these proxies all record a slightly different aspect of the

EASM, still limiting a comprehensive understanding of the drivers and amount of past monsoon precipitation change.

Glycerol dialkyl glycerol tetraethers (GDGTs; Supplementary Fig. S1) are a suite of membrane-spanning lipids of archaea

and bacteria, and have provided another means to reconstruct moisture availability on the CLP. For example, by comparing

the abundance of archaeal isoprenoid GDGTs (isoGDGTs) to that of branched GDGTs (brGDGTs) produced by soil bacteria

in loess-paleosol sequences, where values > 0.5 for this ratio ($R_{i/b}$) indicate arid conditions (Xie et al., 2012; Yang et al.,

2014). The threshold for this ratio is based on the assumption that ammonia oxidation is enhanced in alkaline (pH > 7.5) and

arid soils, like loess, which promotes the growth of ammonia-oxidizing Nitrososphaerota (formerly Thaumarchaeota) that

produce isoGDGTs (Xie et al., 2012). Similarly, the Branched and Isoprenoid Tetraether (BIT) index, representing the

relative abundance of brGDGTs to the isoGDGT crenarchaeol (Hopmans et al., 2004), i.e., basically an inverse of the $R_{i/b}$,

has been linked to soil moisture availability globally (Wang et al., 2013; Dirghangi et al., 2013). Application of the $R_{i/b}$ and

the BIT index to the Weinan section on the southern CLP has revealed sharp peaks in proxy values during glacial

terminations, which have been interpreted as the occurrence of megadroughts, likely caused by El Niño-like conditions in the

tropical Pacific reducing the moisture transport to the Asian continent (Tang et al., 2017). Although of crucial value, these

proxies also do not reflect quantitative changes in monsoon precipitation on the CLP.

Notably, the structural diversity of brGDGTs has been linked to environmental conditions. For example, brGDGTs can vary

in the number of methyl branches attached to the alkyl backbone, which has been linked to growing season temperature

(Weijers et al., 2007; Peterse et al., 2012; De Jonge et al., 2014b; Dearing Crampton-Flood et al., 2020). This relationship

has resulted in absolute temperature reconstructions for the EAM region (Peterse et al., 2011, 2014; Gao et al., 2012; Jia et

al., 2013; Yang et al., 2014; Lu et al., 2019, 2022; Guo et al., 2024) that currently go as far as 3 Ma back in time (Lu et al.,

2022), indicating that brGDGTs are generally well preserved in loess. In addition, brGDGTs can also vary in the number of

internal cyclization (degree of cyclization, DC; Sinninghe Damsté et al., 2009; Baxter et al., 2019) and the position of the

methylation, which can vary between position C-5 and C-6 (De Jonge et al., 2013). Both the DC and the relative abundance

of 6-methyl brGDGT isomers, quantified in the isomer ratio (IR; De Jonge et al., 2014a) are positively related to pH in

modern surface soils. Given that soil pH is typically inversely related to precipitation as a result of increased leaching that

lowers the soil pH with more precipitation (Slessarev et al., 2016), this relationship offers a basis for inferring past changes

in precipitation using soil pH-related proxies. An early study applied this concept to the Mangshan section in the southeast of

the CLP, where brGDGT-based pH was inferred to reflect the precession-forced pacing of monsoon precipitation (Peterse et

al., 2014). However, their reconstructed pH values are higher during the wetter interglacials than during the more arid

glacials, which is opposite to what is generally expected and could thus not be quantitatively translated into realistic

precipitation changes. In addition, this study predates the discovery of 5- and 6-methyl brGDGT isomers, which were not

separated with the chromatography method used at the time, but seem to have a different pH dependency in alkaline soils

(e.g., Guo et al., 2022).





Here, we examine the potential of the pH-sensitive ratios DC and IR as proxies for past monsoon precipitation in a ~130-kyr-
long loess-paleosol sequence at Yuanbao, located on the western edge of the CLP. The proximity of Yuanbao to the main
dust source has led to high deposition rates, allowing for the generation of a high-resolution record (Chen et al., 1997; Sun et
al., 2021). The hydrogen isotope composition of plant waxes ($\delta^2H_{wax}$) from the same section has demonstrated the sensitivity
of Yuanbao records to orbital- and millennial-scale changes in EASM associated precipitation dynamics (Fuchs et al., 2023),
and provides context for the assessment of the DC and IR as quantitative records of past monsoon precipitation on the CLP.

**2 Methods**

**2.1 Study site and sampling**

The Yuanbao section (35.63° N, 103.17° E, 2177 m above sea level) is situated near Linxia city on the western edge of the
CLP (Fig. 1). Mean annual precipitation (MAP) is 500 mm with 80% occurring from May to September (China
Meteorological Administration, http://www.cma.gov.cn, Supplementary Fig. S2). Samples (5 cm intervals) were collected
from the Yuanbao section in August 2019 from a 6 m-deep pit and a partly overlapping 38 m-deep outcrop. Together, they
cover the S0 (Holocene paleosol), L1 (last glacial loess) and S1 (last interglacial paleosol). The chronology of this section is
established by Fuchs et al. (2023). In summary, the GS and MagSus records from Yuanbao were aligned to those from a
nearby drilling core (Guo et al., 2021), that is in turn linked to the benthic $\delta^{18}O$ record, supported by optically stimulated
luminescence (OSL) data (Lai and Wintle, 2006; Lai et al., 2007). In addition, Heinrich stadials (HS) in the GS records were
tied to the absolute uranium-thorium dated speleothem $\delta^{18}O$ record (Cheng et al., 2016). The chronology indicates that the
loess-paleosol sequence of Yuanbao in this study covers the last ~130 kyr, corresponding with a sedimentation rate of 17.8
cm ka$^{-1}$ for S0, 39.3 cm ka$^{-1}$ for L1 and 21.9 cm ka$^{-1}$ for S1 on average.



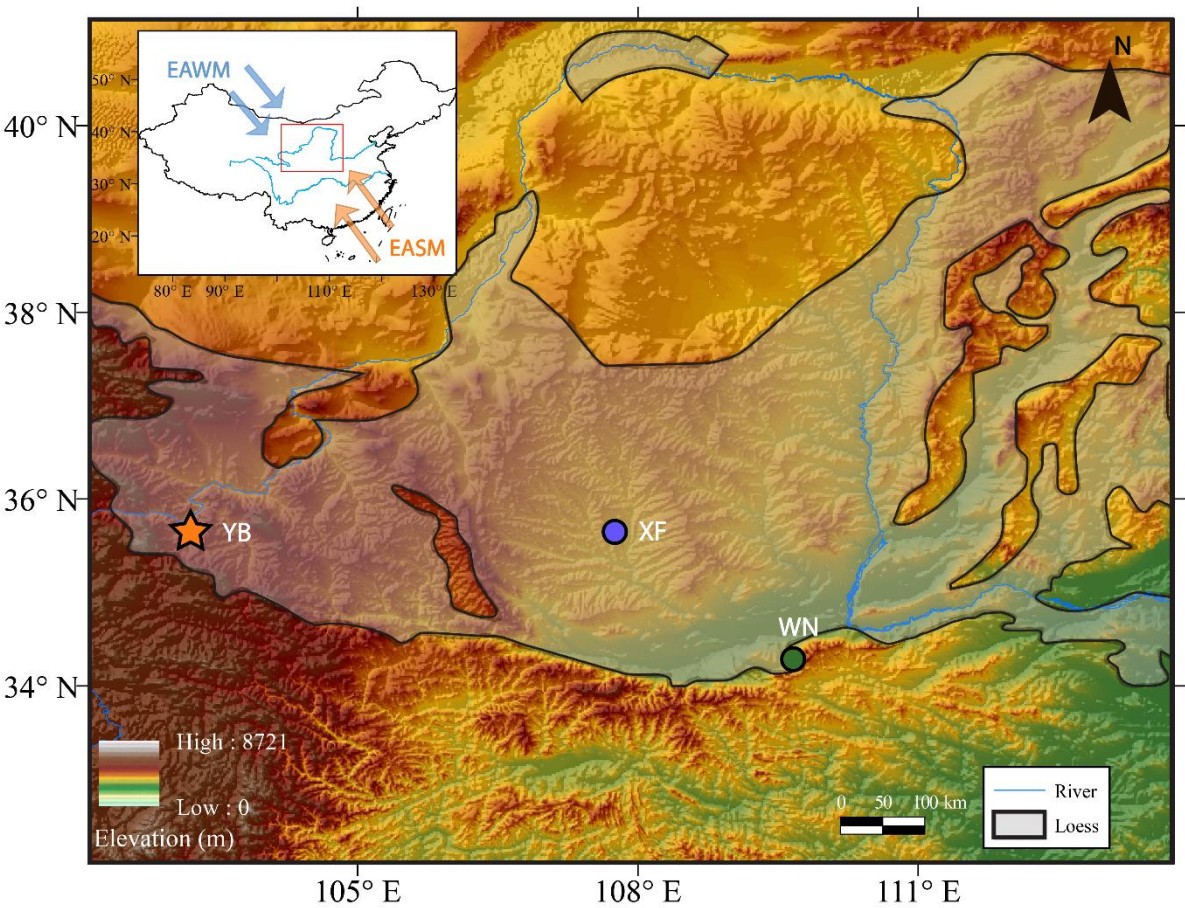

**Figure 1: Map of the Chinese Loess Plateau indicating the study site (Yuanbao, YB, orange star) and relevant wind patterns. The orange and blue arrows represent East Asian Summer Monsoon (EASM) and East Asian Winter Monsoon (EAWM), respectively. Loess-paleosol sequences mentioned in the text are also indicated (Weinan, WN, green circle, and Xifeng, XF, purple circle).**

### 2.2 GDGT extraction and analysis

To avoid anthropogenic perturbation, the upper 30 cm of both the pit and outcrop were excluded from analysis. In total, 580 samples were analysed for GDGTs corresponding with an average resolution of our record of ~220 years. For each depth, ~30–60 g of freeze dried and homogenized loess was extracted with dichloromethane (DCM):methanol (MeOH) (9:1, v/v) using a Milestone ETHOS X (MEX) microwave extractor at 70 °C to obtain a total lipid extract (TLE). The TLEs were filtered using pre-extracted filter paper (Whatman grade 42 Ashless Filter Paper, 55 mm diameter) to remove remaining sediment, and dried under a $N_2$ stream. Prior to analysis for GDGTs, a known amount of $C_{46}$ glycerol trialkyl glycerol tetraethers (GTGT) internal standard was added to the TLEs (Huguet et al., 2006). The TLEs were separated into apolar and





polar fractions by passing them over an activated $Al_2O_3$ column using hexane:DCM (9:1, v/v) and DCM:MeOH (1:1, v/v), respectively. The polar fraction containing brGDGTs was evaporated to dryness under a gentle $N_2$ stream. After this, the samples were re-dissolved in a hexane:isopropanol (99:1, v/v) mixture, and passed over a 0.45 μm polytetrafluoroethylene (PTFE) filter. The GDGTs were injected on an Agilent 1260 Infinity ultra high performance liquid chromatography

(UHPLC) coupled to an Agilent 6130 single quadrupole mass spectrometer (MS) with settings according to Hopmans et al. (2016). GDGTs were detected using selected ion monitoring (SIM). Quantitation was achieved by peak area integration of the $[M+H]^+$ ions in Chemstation software B.04.03.

### 2.3 Proxy calculation

GDGT proxies were calculated according to the equations listed below, where the Roman numerals refer to the molecular

structures shown in Supplementary Fig. S1. Fractional abundances (FA) of GDGTs are indicated by squared brackets. The 6-methyl brGDGTs are denoted with a prime symbol (′).

The degree of cyclization (DC) and the isomer ratio (IR) were calculated according to Baxter et al. (2019) and (De Jonge et al., 2014a), respectively:

DC = ([Ib] + 2*[Ic] + [IIb] + [IIb′])/([Ia] + [Ib] + [Ic] + [IIa] + [IIa′] + [IIb] + [IIb′])                    (1)

IR = ([IIa′] + [IIb′] + [IIc′] + [IIIa′] + [IIIb′] + [IIIc′])/([IIa] + [IIa′] + [IIb] + [IIb′] + [IIc] + [IIc′] + [IIIa] + [IIIa′] + [IIIb] + [IIIb′] + [IIIc] + [IIIc′])                    (2)

### 2.4 Data analysis and visualization

Data visualization was performed in R software (version 4.2.2) (R Core Team, 2022). Scatter plots and line plots were generated with package "ggplot2". Spearman's correlation was used and considered significant at a level of $p < 0.05$. The

DC-MAP calibration was established using Deming regression with package "mcr" (Method Comparison Regression) following the Rscript published by Naafs et al. (2017), which accounts for error in both the proxy (i.e., DC) and environmental parameter (i.e., MAP). To calculate the ratio of variances, the standard deviation of measured MAP of 120 mm and an assumed standard deviation of DC (0.05) were used. The calibration uncertainty is determined by the residuals of Deming regression. Bandpass filter and spectral analysis were conducted using Acycle (version 2.3) (Li et al., 2019). The

records were standardized, detrended (three order polynomial fit), and interpolated to obtain constant time steps before spectral analysis using the periodogram function with default settings. Cross-spectral analysis between precession and East Asian monsoon proxies, ice volume and greenhouse gas were performed with the Blackman-Tukey approach using arand-master software and interpolation of the records at a constant time step of 1 ka. The following parameters were used to optimize variance properties of spectrum estimates: number of lags = 40 (~1/3 length of record) and samples per analysis =

120. The coherency spectra are compared at the 80% non-zero coherency level.



## 3 Results







**Figure 2: Overview of brGDGT- and loess-based proxy records for the past 130 kyr at Yuanbao. (A) Degree of cyclization (DC) of brGDGTs and ice-corrected $\delta^2H_{wax}$ based on plant waxes in the same lipid extracts (Fuchs et al., 2023). (B) Bandpass filter of DC, dark orange and light orange curve indicate the 41-kyr and 23-kyr band-pass filters, respectively. (C) Isomer ratio (IR) of brGDGTs. (D) Magnetic Susceptibility (MagSus) and Grain Size (GS) (Fuchs et al., 2023). (E) Northern Hemisphere Summer Insolation at 35°N (NHSI in June, July and August; Berger et al., 2010) and the composite speleothem oxygen isotope ($\delta^{18}O$) record (Cheng et al., 2016). Grey intervals (~23–21 ka) in brGDGT-related records (DC and IR) indicate the transition from the outcrop to the pit and are not considered in the interpretation of the records. Light blue bars in the background indicate Marine Isotope Stages (MIS), and light grey bars indicate Heinrich stadials (HS). Brown and yellow rectangles above the x-axis represent paleosol (S) and loess (L) layers, respectively.**

Downcore changes in the distribution of brGDGTs result in IR values varying between 0.2 and 0.8 over the past 130 kyr, where IR values are generally low during MIS5, but show a rapid increase from 0.2 to 0.6 within a ~10 kyr interval over the transition from MIS5a into MIS4 (from ~73 to 63 ka), after which the IR increases to maximum 0.8 during the Holocene (Fig. 2C). DC values vary between 0.2 and 0.7 over the past 130 kyr, and are high (0.4–0.6) during interglacials (i.e., MIS1 and MIS5e) and low (0.2–0.4) during glacials (i.e., MIS2 and MIS4) (Fig. 2A). Notably, millennial-scale variations in the DC record match the variability recorded by GS and $\delta^2H_{wax}$ in the same section, with prominent dry periods associated with the HS (Fig. 2A and D). Bandpass filters (Fig. 2B) and spectral analysis (Fig. 3) of the DC record indicate the presence of dominant 41-kyr and 23-kyr cycles.

Both brGDGT-based proxy records show abrupt changes between ~23 and 21 ka, corresponding to the transition from the outcrop to the pit (indicated in grey in Fig. 2A and C). Although this transition does not stand out in the conventional loess proxy records (MagSus and GS) derived from the same material (Fig. 2D), or the $\delta^2H_{wax}$ record obtained from the same lipid extracts (Fig. 2A), this interval is not considered in our interpretation of the brGDGT proxy records.







185





**Figure 3: Spectra of time series of proxy records from the Yuanbao loess section and cave speleothems in southeast China.** Spectral analysis of (A) Degree of cyclization (DC), (B) Ice-corrected $\delta^2H_{wax}$ record, (C) magnetic susceptibility (MagSus), and (D) grain size (GS) from Yuanbao, and (E) speleothem $\delta^{18}O$. All spectral analysis span to 132 ka. Red line represents the 99% significance level bending power law (BPL). Vertical grey bars indicate primary orbital periods of the 23-, 41-, and 100-kyr cycles.

## 4 Discussion

### 4.1 Downcore changes in brGDGT-related records at Yuanbao

Although the 6-methyl brGDGTs respond to soil pH in the global surface soil dataset (De Jonge et al., 2014b; Raberg et al., 2022), and thus presumably also to precipitation amount, the IR record for Yuanbao does not exhibit glacial-interglacial or precession cycles that are visible in other precipitation-related proxy records for this site, such as MagSus and $\delta^2H_{wax}$ (Fig. 2A, C and D). By contrast, the DC record shows a trend matching that of the $\delta^2H_{wax}$ record (Fig. 2A), including a precession signal and millennial-scale climate variations, suggesting that the DC may in fact respond to precipitation changes. Indeed, a previous study has empirically linked the cyclization of brGDGTs in modern Chinese alkaline soils to soil pH and MAP (Wang et al., 2014). However, this study also predates the discovery of 6-methyl brGDGT isomers and thus comprises both the DC and IR. The different trends in the IR and DC records for Yuanbao clearly indicate a distinct pH dependency of these indices in loess (Fig. 2A and C). Based on our current understanding of 5- and 6-methyl brGDGT distributions in modern soils, the 0.4 increase in IR within ~10 kyr (from ~73 to 63 ka) would correspond to a change in soil pH of ~2 units (Raberg et al., 2022). Such a large shift in pH can only be explained by a major alteration in source material or hydrological conditions. Although the other proxy records for this site (increased GS and $\delta^2H_{wax}$ values, decreased DC and MagSus values) indicate a hydrological transition towards more arid conditions, this event only last for ~3 kyr, after which the records indicate sustained dry conditions until ~60 ka (Fig. 2). The same event is also present in the speleothem $\delta^{18}O$ record where is referred to as Chinese-Stadial 20, representing the weakest EASM during the last glacial (Du et al., 2019; Zhang et al., 2017). Even though this shift to cold and dry conditions is shorter than the shift in IR, this hydroclimatic change may have triggered a microbial community response on the CLP.

In contrast, an explanation for the relationship between the DC and precipitation in loess-paleosol sequences is more straightforward. Recent studies have indicated that the production of brGDGTs with cyclopentane moieties is promoted over that of non-cyclic brGDGTs by the availability of exchangeable calcium ions ($Ca^{2+}$) (De Jonge et al., 2021; Halffman et al., 2022). Since loess is generally rich in $CaCO_3$, it is likely that more $Ca^{2+}$ will be available during periods with enhanced precipitation, which will facilitate the dissolution of $CaCO_3$. Indeed, the DC in the Yuanbao section correlates well with variations in the Ca/Ti record that reflects precipitation-induced weathering intensity in the nearby drill core (Guo et al., 2021). Assuming that brGDGTs respond to the $Ca^{2+}$ availability at the time they are produced, periods with increased precipitation on the CLP will lead to an increase in the DC. Together with the trend in the DC record, this mechanism supports the potential of using DC as a precipitation proxy on the CLP.



## 4.2 Quantification of past monsoon precipitation amount

To assess the ability of the DC to record changes in precipitation amount, we determined the relationship of the DC with soil
220  pH and MAP in the global surface soil dataset (Fig. 4). The DC shows different relationships with soil pH in more acidic and
more alkaline soils. Specifically, the DC is (exponentially) positively related with pH in soils with pH < 7 ($r^2 = 0.38$, $p <$
0.01, Fig. 4A), while it is negatively, and only weakly related to the soil pH in alkaline soils with pH > 7 ($r^2 = 0.059$, $p <$
0.01, Fig. 4A), as observed before in modern Chinese soils (Xie et al., 2012). A recent study comprising mid- and high-
latitude soils demonstrated that the change in the direction of this relationship coincided with a shift in the soil bacterial
225  community (De Jonge et al., 2021). This suggests that brGDGTs in high and low pH soils may have different producers that
respond to environmental changes differently. Notably, the IR based on the same dataset does not show this change in
relationship with soil pH (Fig. 4B), suggesting that DC and IR record different environmental parameters, specifically in
alkaline soils.







**Figure 4: Spearman's correlation of measured soil pH with (A) degree of cyclization (DC) and (B) isomer ratio (IR), and mean annual precipitation (MAP) with (C) DC and (D) IR. The green, purple, and yellow symbols indicate soils with pH < 7 ($n$ = 833 in panels A, B, $n$ = 323 in panels C, D), soils with pH > 7 ($n$ = 510 in panels A, B, $n$ = 315 in panels C, D) in the global soil dataset (Raberg et al., 2022), and modern soils from the Chinese Loess Plateau (CLP; $n$ = 46 in panels A, B, $n$ = 35 in panels C, D; Wang et al., 2020), respectively. The linear regressions are represented by solid lines and 95% confidence intervals are indicated by the shaded area. Panel (E) and (F) shows scatterplot of DC versus measured MAP of modern soils from the CLP [Eq. (3)] and the residuals for Deming regression (i.e., offset between the measured and calculated MAP values based on [Eq. (3)]), RMSE, root mean square error. The locations of the datapoints from the CLP are in Supplementary Fig. S3. Panels (C), (D), (E) and (F) only include soils for which MAP data were available in the original studies.**

Although soil pH and MAP are related on a global scale, only the DC in soils with pH > 7 shows correlations with MAP ($r^2$ = 0.37, $p$ < 0.01, Fig. 4C). Notably, all modern soils from the CLP fall within this cluster, supporting the potential to establish the DC as quantitative proxy for monsoon precipitation in this area. Based on the relationship between the DC for modern soils from the CLP and MAP from meteorological stations, the following transfer function can be derived using Deming regression, which accounts for errors in both proxy and environmental variable data (Fig. 4E and F):

$$MAP = 1112 * DC + 248 \quad (n = 35, r^2 = 0.77, RMSE = 71 \text{ mm}, p < 0.01) \quad \quad (3)$$

Even though we used Deming regression to avoid regression dilution that can appear in linear regressions, there still is a trend in the residuals, suggesting that other factors may influence the DC, in addition to MAP (Fig. 4F). Nevertheless, the strong correlation of the regression ($r^2$ = 0.77) indicates that MAP is the main control of DC on the CLP. Based on the range in the residuals, the uncertainty on MAP estimates is ± 125 mm (Fig. 4F).

Using this function to translate the Yuanbao DC record into MAP results in values varying between 460 to 1000 mm over the past 130 kyr, where MAP was highest during MIS5e, and lowest during MIS2 (Fig. 5A). Although anthropogenic disturbance of the upper part of the loess section does not allow us to directly compare DC-based MAP with modern precipitation data at Yuanbao, reconstructed MAP matches well with a previous reconstruction of summer precipitation for a nearby loess section based on $\delta^{13}C_{TOC}$ (Rao et al., 2013), assuming that summer (June, July, August) precipitation accounts for ~50% of the MAP as is the case under modern climate conditions (Supplementary Fig. S2). Nonetheless, the range of the DC in surface soils (0.06 to 0.50) does not cover the full range of the DC in the Yuanbao paleorecord (0.19 to 0.68), which may introduce additional uncertainty to reconstructed MAP, especially for periods with a DC > 0.5 (e.g., MIS5e and the Holocene).

To provide more context for the reconstructed MAP at Yuanbao, and to assess the wider applicability of our transfer function, we use the DC to reconstruct MAP at two other sections located on the middle (Xifeng; Lu et al., 2019) and southern (Weinan; Tang et al., 2017) CLP for which brGDGT data are available for the same time interval (Fig. 5B and C). Both MAP records generally show a similar precipitation pattern to that at Yuanbao, albeit in a lower resolution. Hence, millennial-scale events that are reflected in the Yuanbao record are not recorded at Xifeng or Weinan. Still, reconstructed MAP faithfully reflects wetter conditions during interglacials and drier conditions during glacials at both locations, and



265   ranges from 470 to 900 mm at Xifeng and from 660 to 1030 mm at Weinan. These MAP estimates are in the same range as
those based on $^{10}$Be from nearby sections (Zhou et al., 2023; Beck et al., 2018). Considering the uncertainty on the DC-MAP
calibration of ± 125 mm, the records indicate that all sites received approximately the same amount of EASM precipitation
during interglacials i.e., ~850 mm during the Holocene optimum and ~950 mm during MIS5e, but that spatial differences in
precipitation occurred during glacials, where the southern part of the CLP received up to ~250 mm more precipitation than

270   the middle and the western parts of the CLP. These spatial and absolute trends in MAP are in agreement with the presumed
reach of the EASM rain belt to west of the CLP during interglacials, and gradual shifts during glacial-interglacial transitions
(An et al., 2000; Li et al., 2018; Yang et al., 2015).









**Figure 5: The mean annual precipitation (MAP) based on the degree of cyclization (DC) of brGDGTs for different sections among the Chinese Loess Plateau (CLP), i.e., (A) Yuanbao (YB, western CLP), (B) Xifeng (XF, middle CLP), and (C) Weinan (WN, southern CLP), calibration uncertainty is determined by the residuals of Deming regression shown in Fig. 4F. (D) Oxygen isotope composition (δ$^{18}$O) of stacked speleothem records from Hulu/Sanbao/Dongge caves (Cheng et al., 2016). The grey interval (~23–21 ka) in the MAP record for Yuanbao indicates the transition from the outcrop to the pit. Light blue bars in the background indicate Marine Isotope Stages (MIS). Brown and yellow rectangles above the x-axis represent paleosol (S) and loess (L) layers, respectively.**

## 4.3 Implications for EASM precipitation reconstructions

The close resemblance of the DC and δ$^2$H$_{wax}$ records for Yuanbao suggests that the δ$^2$H$_{wax}$ signal largely represents precipitation amount. However, offsets between these two records occur during MIS5, the transition from MIS4 to MIS3, and the last glacial maximum (Fig. 2A). Specifically, the DC record indicates wetter conditions during MIS5e than during MIS5c and MIS5a, while the δ$^2$H$_{wax}$ record shows increasingly depleted values from MIS5e to MIS5a, suggesting more intense EASM precipitation during MIS5a (Fig. 2A). This opposite trend can be explained by warm climate conditions during MIS5e, which would enhance evapotranspiration and local recycling, and result in less negative δ$^2$H$_{wax}$ values (Liu et al., 2019; Sachse et al., 2012; Gat, 1996). Similarly, the DC record indicates dry conditions during the second half of MIS4 and the last glacial maximum, while the δ$^2$H$_{wax}$ record shows more negative values, suggesting that δ$^2$H$_{wax}$ was likely impacted by a change in moisture source during this time, possibly derived from the Westerlies that bring moisture relatively depleted in $^2$H (Fuchs et al., 2022).

Importantly, next to the precession signal that dominates the records of EASM precipitation based on precipitation isotopes (i.e., δ$^2$H$_{wax}$ (Fuchs et al., 2023), speleothem δ$^{18}$O (Cheng et al., 2016)), the DC record from Yuanbao additionally shows a clear obliquity signal (Fig. 3A, B and E, Supplementary Fig. S4). This could be introduced by the longer growing season of soil microbes that produce brGDGTs under generally warm and wet conditions compared to that of leaf wax synthesis incorporating precipitation $^2$H by plants. Similarly, reduced microbial activity in winters may also explain the absence of a clear glacial-interglacial cycle that appears in MagSus and GS record from the same section (Fig. 3A, C and D; Fuchs et al., 2023), although the manifestation of this cycle is likely also restricted by the length of our record (~130 kyr). The obliquity signal in the DC record suggests that while moisture source and intensity of the EASM follow precession, the amount of precipitation at Yuanbao responds to the summer insolation at high-latitude regions, which contains a stronger obliquity signal than at lower latitudes. Reconstructed precipitation amount based on microcodium Sr/Ca ratios from the middle CLP similarly show an obliquity cycle over the last 1500 kyr (Li et al., 2017). However, the dominant orbital forcings driving EASM precipitation vary spatially across the CLP and also between different proxies. For example, precipitation amount reconstructed for the middle (Xifeng; Zhou et al., 2023) and southern (Baoji, Beck et al., 2018) part of the CLP based on $^{10}$Be show precession and glacial-interglacial cycles. Although spatial differences in loess deposition rates may influence the expression of obliquity and precession cycles in different proxy records (e.g., Ao et al., 2024), the offsets also suggest that





these proxies are sensitive to different aspects of EASM precipitation, integrate a different part of the year, and/or that the amount of monsoon precipitation is increasingly influenced by high-latitude insolation towards the northwestern CLP.

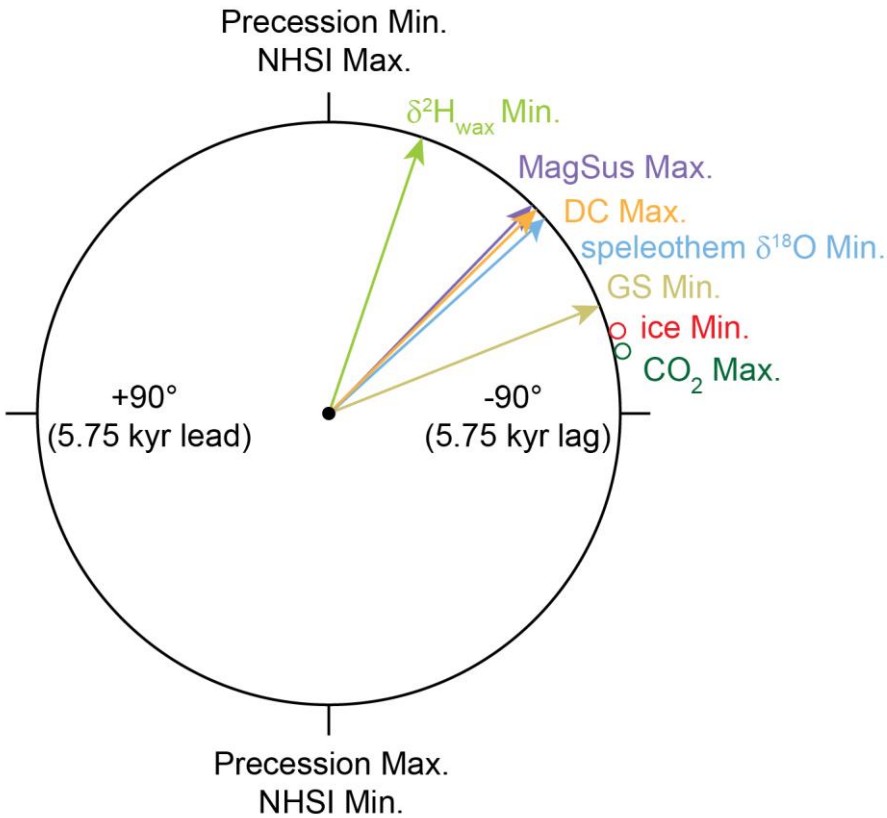


**Figure 6: Phase wheel comparing proxy records at Yuanbao, oxygen isotope composition ($\delta^{18}O$) of cave speleothems (Cheng et al., 2016), global ice volume (Lisiecki and Raymo, 2005) and $CO_2$ (Bereiter et al., 2015) to precession (Berger et al., 2010). Zero on the precession phase wheel is set at precession minimum when the Earth is at perihelion and the North Pole is pointed directly at the Sun, which corresponds to the Northern Hemisphere Summer Insolation (NHSI) maximum. Proxy records from Yuanbao:**
**hydrogen isotope composition of plant waxes ($\delta^2H_{wax}$), loess proxies (magnetic susceptibility, MagSus and grain size, GS), degree of cyclization (DC) of brGDGTs.**

Despite the differences between proxies and sections across the CLP, the spectral analysis of proxy records for Yuanbao and speleothem $\delta^{18}O$ show a consistent precession signal (Fig. 3). We subsequently performed cross-spectral analyses to
compare the timing of EAM records with the precession extremes. The DC record, as an indicator of precipitation amount, lags the precession minimum (i.e., NHSI maximum) by ~3 kyr. Similarly, the MagSus, $\delta^2H_{wax}$ and $\delta^{18}O$ are coherent with precession and lag the precession minimum by ~3 kyr, 1 kyr and 3 kyr, respectively, indicating that they are relatively in phase (Fig. 6). Despite the arguments against interpreting speleothem $\delta^{18}O$ as an EASM precipitation signal (e.g., Clemens et al., 2010), the independent $\delta^2H_{wax}$ record for the Yuanbao section is also dominated by a precession cycle, and in phase with

the cave records, suggesting that both sites have recorded the same moisture source (Fuchs et al., 2023). Although $\delta^2H_{wax}$ may be influenced by the same confounding factors as the cave records, the DC-based MAP record is independent of precipitation isotope composition, and yet it is in phase with speleothem $\delta^{18}O$ and $\delta^2H_{wax}$ at the precession cycle (Fig. 6). Thus, our record supports the hypothesis that NHSI exerts a stronger influence on EASM precipitation than ice volume and greenhouse gas.

## 330   5 Conclusions

The downcore distribution of brGDGTs at the Yuanbao section on the western CLP reveals that absolute changes in monsoon precipitation over the past ~130 kyr are recorded by the degree of cyclization (DC) of the brGDGTs, based on their relationship with exchangeable $Ca^{2+}$ that becomes available upon the dissolution of $CaCO_3$ during periods with enhanced precipitation. The DC record aligns with the orbital- and millennial-scale variations in EASM precipitation recorded by
speleothem $\delta^{18}O$ and the $\delta^2H_{wax}$ from the same section. Using the modern DC-MAP relationship to quantify past MAP across the CLP reveals that MAP consistently varies over glacial-interglacial timescales, but that differences in precipitation amount are most pronounced during glacials, when the middle and western CLP experience relatively more arid conditions compared to the southern CLP. The in-phase relationship of the DC with $\delta^2H_{wax}$ from the same section, as well as with speleothem $\delta^{18}O$ on the precession band suggests that the supposed response of monsoon precipitation to NHSI is not an
artefact of confounding factors on proxies recording precipitation isotopes, but that NHSI is a direct forcing of precipitation amount.

**Data availability**

All data have been submitted to PANGEA database, a DOI link will be provided when it is available.

**Author Contribution**

F.P., M.Z. and Y.S. designed the study, F.P. and Y.S. collected the sample material. J.G. and L.F. conducted the biomarker analysis. J.G. interpreted and visualized the data under supervision of F.P., M.Z. and S.Y., J.G. wrote the paper with input from all co-authors.

**Competing interests**

The authors declare that they have no conflict of interest.



**Acknowledgements**

Klaas Nierop and Desmond Eefting in the GeoLab at Utrecht University are acknowledged for technical support. We thank Hao Lu for providing the map, Shivam Mishra for helping with data generation, and Zhipeng Wu, Fei Guo for helping with wavelet and cross-spectral analysis. This work has received funding from the Dutch Research Council (NWO, Vidi grant no. 192.074 to FP) and National Natural Science Foundation of China (NSFC, grant no. 42230514 to YS).

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
