# Peer review of "Towards quantitative reconstruction of past monsoon precipitation based on tetraether membrane lipids in Chinese loess"

_EGUsphere, 2024_

## Author Comment (AC2)

**Referee comment #3:**

Guo et al. generated new records based on branched glycerol dialkyl glycerol tetraethers (brGDGTs) from the Chinese Loess Plateau (CLP) over the last 130 kyrs. The authors found that two pH-sensitive brGDGT-based indices, DC and IR, showed contrasting temporal changes at the same site. After a comparison of the new brGDGT-based records with several published records from the same site, such as another biomarker-based record (ice-corrected $\delta2H_{wax}$), the authors found that DC showed promise as a mean annual precipitation (MAP) proxy. Then, the authors investigated the relationships between brGDGT-based indices (DC and IR) and soil pH and MAP using a global modern soil dataset, as well as a CLP modern soil dataset. The authors found that in alkaline soils, including within the CLP, DC showed a strong correlation with MAP, which enabled the development of a DC-MAP calibration for quantitative MAP reconstructions. The authors then applied their DC-MAP calibration at three sites within the CLP, including the study site, and investigated spatial differences in MAP within the CLP and their changes through time. The authors also did spectral and cross-spectral analyses and found that (i) their DC record showed precession and obliquity signals, contrary to $\delta2H_{wax}$ and $\delta18O_{speleo}$records which only showed the precession signal, and (ii) their DC record was in phase with $\delta2H_{wax}$ and $\delta18O_{speleo}$records at the precession scale. The authors thus concluded that Northern Hemisphere summer insolation was a direct forcing of precipitation amount rather than the result from confounding factors on precipitation records based on isotopes.

As a paleoclimatologist with expertise on GDGT-based proxies, I have read this manuscript with interest for several reasons. First, the authors tackled a topical and important subject, namely the understanding of the East Asian Monsoon which also has its own controversies and subjects of debate, as the authors stated in the Introduction. Second, this manuscript presents an interesting use of a brGDGT-based proxy, namely DC as a (quantitative) MAP

proxy, as brGDGTs are classically used for quantitative reconstructions of land temperature and soil pH. Third, the authors reconstructed past MAP changes using a proxy that does not involve hydrogen or oxygen isotopes, contrary to δ2Hwax and δ18Ospeleo, which strengthens the independence of DC relative to δ2Hwax and δ18Ospeleo. Furthermore, I found the manuscript easy to read and well-organized. Overall, this piece of work would be a great addition to the literature and is worth publishing in Climate of the Past.

However, I have several comments, questions, and suggestions for revision, which I detail below.

*Reply: We thank the reviewer for their positive evaluation of our work. Please find our point-by-point response below in italic.*

General comments:

1) Recently, Zhao et al. (2020) introduced a precipitation index (PI) as follows: PI = (Ia + Ib)/(Ia + Ib + IIIa + IIa′ + IIIa′). Like the redefined DC, the PI takes advantage of the improved separation of 5- and 6-methyl brGDGT isomers. Importantly, Zhao et al. (2020) and Zhang et al. (2024) proposed PI-MAP calibrations in cancellous bones and soils, respectively, for brGDGT-based quantitative MAP reconstructions. Accordingly, I would like to know the authors' thoughts concerning the PI. Specifically, the authors may check how well the PI would behave as a (quantitative) MAP proxy compared with e.g., DC and IR in the CLP and, in case of similar trends between DC and PI, which MAP reconstructions the PI would yield in the CLP. However, the authors do not need to switch to the PI, especially if DC has a better motivation and/or shows more meaningful results compared with the PI in the authors' view.

Zhang, T., Han, W., Tian, Q., Zhang, J., Kemp, D. B., Wang, Z., Yan, X., Mai, L., Fang, X., and Ogg, J.: Tectonically controlled establishment of modern-like precipitation patterns in East and central Asia during the early late Miocene, Journal of Geophysical Research: Atmospheres, 129, e2024JD041025, https://doi.org/10.1029/2024JD041025, 2024.

Zhao, J., Huang, Y., Yao, Y., An, Z., Zhu, Y., Lu, H., and Wang, Z.: Calibrating branched GDGTs in bones to temperature and precipitation: application to Alaska chronological sequences, Quaternary Science Reviews, 240, 106371, https://doi.org/10.1016/j.quascirev.2020.106371, 2020.

*Reply: We thank the reviewer for their comments and providing the references. In previous studies, the precipitation index (PI) has shown a positive correlation with precipitation in bones and surface soils from China (Zhao et al., 2020; Zhang et al., 2024). The application of the PI in the Yuanbao section, however, results in a record with higher values during stadials than interstadials during MIS5, which is inconsistent with other hydroclimate proxies from the same section (Fig. 1B), such as MagSus and $\delta^2H_{wax}$. In addition, compared to the DC, the PI fails to capture millennial-scale events during the last glacial period.*

*Upon closer examination, the PI equation combines the degree of methylation (MBT), the position of the methylation (IR), and the degree of cyclization (DC), making it complex to interpret which membrane adjustment is driving the changes in the PI record. Specifically, having the tetramethylated brGDGTs in the numerator and the penta- and hexamethylated brGDGTs in denominator resembles the structure of the MBT'$_{5Me}$, which is linked to temperature -and not precipitation- in the global soil dataset (De Jonge et al., 2014). In addition, the PI was initially developed based on distributions of brGDGTs in bones, where bacterial community compositions might differ from those in (arid and alkaline) soils and loess and paleosol materials.*

*For these reasons, we have decided not to include the PI in our revised manuscript.*

*Nevertheless, it would be interesting to test the PI in different types of material in future*

*work.*

[Figure]

**Fig. 1** *Biomarker-based records for the past 130 kyr at Yuanbao. (A) Degree of cyclization (DC) of brGDGTs and ice-corrected δ²Hwax based on plant waxes in the same lipid extracts (Fuchs et al., 2023). (B) brGDGT-based precipitation index (PI) (Zhao et al., 2020). (C)*

*isoGDGT-based mean monthly precipitation (MMP) record (De Jonge et al., 2024). (**D**) NHSI at 35°N (Berger et al., 2010) and the composite speleothem oxygen isotope ($\delta^{18}O$) record (Cheng et al., 2016). Dark grey intervals (~23–21 ka) in brGDGT-related records (DC and PI) indicate the transition from the outcrop to the pit and are not considered in the interpretation of the records.*

2) Even more recently, De Jonge et al. (2024a) proposed another GDGT-based proxy which may track precipitation changes, specifically mean monthly precipitation (MMP) changes: MMP = (isoGDGT-1 + isoGDGT-3)/(isoGDGT-1 + cren). I recognize that the involved GDGTs are isoGDGTs rather than brGDGTs and that the alternative GDGT-based index would likely yield uncertain reconstructions as well (see the [Eq. 14] versus MMP plot in Supp. Fig. 8 in De Jonge et al., 2024a). Nevertheless, provided that isoGDGTs are abundant enough to yield peak areas above quantification limit, I feel that the authors may consider this isoGDGT-based ratio as well.

De Jonge, C., Guo, J., Hällberg, P., Griepentrog, M., Rifai, H., Richter, A., Ramirez, E., Zhang, X., Smittenberg, R. H., Peterse, F., Boeckx, P., and Dercon, G.: The impact of soil chemistry, moisture and temperature on branched and isoprenoid GDGTs in soils: a study using six globally distributed elevation transects, Organic Geochemistry, 187, 104706, https://doi.org/10.1016/j.orggeochem.2023.104706, 2024a.

*Reply: We thank the reviewer for highlighting this isoGDGT-based precipitation proxy. We have applied the proxy to our loess section (Fig. 1C). However, the resulting record does not show a trend that could be linked to any climate events, regionally or globally. Although the work of De Jonge et al., (2024) suggests that isoGDGTs in soils have potential as a proxy for precipitation reconstructions, the drivers of isoGDGT distributions in arid and alkaline soils, like loess, remain elusive. At least, the mean monthly precipitation (MMP) record we obtained for Yuanbao suggests that precipitation may not be the dominant control. We have therefore decided not to include this record into our revised manuscript.*

Detailed comments:

Main text

Line 11: In "Chinese Loess Plateau", "loess" is not capitalized in the abstract, but is in line 40 in the Introduction.

*Reply: We will correct this in the revised manuscript.*

Line 48: Which paper by Guo et al. (2022) is cited here? The one published in Geology (Guo et al., 2022a)?

*Reply: It is the Geology one indeed. We will make it clear in the revised manuscript.*

Line 98: Which paper by Guo et al. (2022) is cited here? The one published in Organic Geochemistry (Guo et al., 2022b)?

*Reply: We thank the reviewer for their careful check, this is the one published in Organic Geochemistry. We will make it clear in the revised manuscript.*

Fig. 1 (also Fig. S3): Readers may find it hard to read the coordinate labels for the inset which shows the relevant wind patterns: the authors should consider changing the color from black to white, as they did for other labels within the larger map, and/or increasing the font size.

*Reply: We will modify this accordingly in the revised manuscript.*

Lines 142–143: Replace "(De Jonge et al., 2014a)" with "De Jonge et al. (2014a)".

*Reply: We will replace this in the revised manuscript.*

Fig. 2: For panel A, the authors should pick a color pair different from the current green-orange one for the sake of accessibility to color-blind readers. For panel B, the authors could consider picking a color pair with a stronger contrast in terms of hue and/or lightness for the sake of readability.

*Reply: We will change this in the revised manuscript.*

Line 204: Replace "this event only last" with "this event only lasts".

*Reply: We will correct this in the revised manuscript.*

Fig. 4: For panels A–D, the authors should consider revising the colors to avoid the green-orange confusion for color-blind readers. Alternatively, the authors should distinguish the CLP datapoints from the other ones using a different symbol type, for instance with squares, diamonds, or triangles rather than circles. If the authors pick the second option, then the change in symbol type for CLP should be reflected in panels E and F as well.

*Reply: We will modify this accordingly in the revised manuscript.*

Lines 247–248: The r2 value represents the percentage of variance explained by the regression, not the correlation strength which is represented by the r value.

*Reply: We will change this in the revised manuscript.*

Lines 255–258: This is an important and welcome remark.

*Reply: We thank the reviewer for their positive feedback.*

Line 294: "(i.e., δ2Hwax (Fuchs et al., 2023), speleothem δ18O (Cheng et al., 2016))": A few parentheses should be removed so that only a pair of parentheses remains.

*Reply: We have carefully checked and will make sure they are well in order in the revised manuscript.*

References: Could the authors recheck their reference list? The formatting appears a bit suboptimal at places, for instance in lines 367–368 (Baxter et al., 2019) where I spotted a "ScienceDirect" which appears out of place there, as well as in lines 537–539 (Wang et al., 2001) where I spotted an unexpected "(80-. )." just after the journal name.

*Reply: We thank the reviewer for their careful check. We will carefully go through the reference list and correct the formatting.*

Supplementary Figures

Fig. S1: It would be great if the authors could write the m/z values of [M+H]+ ions with at least one decimal place rather than as integer values. Otherwise, researchers who would examine GDGTs for the first time may fail to do optimal GDGT analyses for the reasons discussed by Davtian et al. (2018) and partly reminded by De Jonge et al. (2024b).

Davtian, N., Bard, E., Ménot, G., and Fagault, Y.: The importance of mass accuracy in selected ion monitoring analysis of branched and isoprenoid tetraethers, Organic Geochemistry, 118, 58–62, https://doi.org/10.1016/j.orggeochem.2018.01.007, 2018.

De Jonge, C., Peterse, F., Nierop, K. G. J., Blattmann, T. M., Alexandre, M., Ansanay-Alex, S., Austin, T., Babin, M., Bard, E., Bauersachs, T., Blewett, J., Boehman, B., Castañeda, I. S., Chen, J., Conti, M. L. G., Contreras, S., Cordes, J., Davtian, N., van Dongen, B., Duncan, B., Elling, F. J., Galy, V., Gao, S., Hefter, J., Hinrichs, K.-U., Helling, M. R., Hoorweg, M., Hopmans, E., Hou, J., Huang, Y., Huguet, A., Jia, G., Karger, C., Keely, B. J., Kusch, S., Li, H., Liang, J., Lipp, J. S., Liu, W., Lu, H., Mangelsdorf, K., Manners, H., Martinez Garcia, A., Menot, G., Mollenhauer, G., Naafs, B. D. A., Naeher, S., O'Connor, L. K., Pearce, E. M., Pearson, A., Rao, Z., Rodrigo-Gámiz, M., Rosendahl, C., Rostek, F., Bao, R., Sanyal, P., Schubotz, F., Scott, W., Sen, R., Sluijs, A., Smittenberg, R., Stefanescu, I., Sun, J., Sutton, P., Tierney, J., Tejos, E., Villanueva, J., Wang, H., Werne, J., Yamamoto, M., Yang, H., and Zhou, A.: Interlaboratory comparison of branched GDGT temperature and pH proxies using soils and lipid extracts, Geochemistry, Geophysics, Geosystems, 25, e2024GC011583, https://doi.org/10.1029/2024GC011583, 2024b.

*Reply: We thank the reviewer for their careful review and detailed suggestions. We will add this important information to the Method and supplementary figure in the revised manuscript to avoid any potential misleading.*

**References**

*Berger, A., Loutre, M.-F., and Yin, Q.: Total irradiation during any time interval of the year using elliptic integrals, Quat. Sci. Rev., 29, 1968–1982, https://doi.org/10.1016/j.quascirev.2010.05.007, 2010.*

Cheng, H., Edwards, R. L., Sinha, A., Spötl, C., Yi, L., Chen, S., Kelly, M., Kathayat, G., Wang, X., Li, X., Kong, X., Wang, Y., Ning, Y., and Zhang, H.: The Asian monsoon over the past 640,000 years and ice age terminations, Nature, 534, 640–646, https://doi.org/10.1038/nature18591, 2016.

Fuchs, L., Guo, J., Schefuß, E., Sun, Y., Guo, F., Ziegler, M., and Peterse, F.: Isotopic and magnetic proxies are good indicators of millennial-scale variability of the East Asian monsoon, Commun. Earth Environ., 4, 425, https://doi.org/10.1038/s43247-023-01090-z, 2023.

De Jonge, C., Hopmans, E. C., Zell, C. I., Kim, J.-H., Schouten, S., and Sinninghe Damsté, J. S.: Occurrence and abundance of 6-methyl branched glycerol dialkyl glycerol tetraethers in soils: Implications for palaeoclimate reconstruction, Geochim. Cosmochim. Acta, 141, 97–112, https://doi.org/10.1016/j.gca.2014.06.013, 2014.

De Jonge, C., Guo, J., Hällberg, P., Griepentrog, M., Rifai, H., Richter, A., Ramirez, E., Zhang, X., Smittenberg, R. H., Peterse, F., Boeckx, P., and Dercon, G.: The impact of soil chemistry, moisture and temperature on branched and isoprenoid GDGTs in soils: A study using six globally distributed elevation transects, Org. Geochem., 187, 104706, https://doi.org/10.1016/j.orggeochem.2023.104706, 2024.

Zhang, T., Han, W., Tian, Q., Zhang, J., Kemp, D. B., Wang, Z., Yan, X., Mai, L., Fang, X., and Ogg, J.: Tectonically Controlled Establishment of Modern-Like Precipitation Patterns in East and Central Asia During the Early Late Miocene, J. Geophys. Res. Atmos., 129, 1–14, https://doi.org/10.1029/2024JD041025, 2024.

Zhao, J., Huang, Y., Yao, Y., An, Z., Zhu, Y., Lu, H., and Wang, Z.: Calibrating branched GDGTs in bones to temperature and precipitation: Application to Alaska chronological sequences, Quat. Sci. Rev., 240, 106371, https://doi.org/10.1016/j.quascirev.2020.106371,

*2020.*

---

## Author Response (AR1)

Dear editor, dear Prof. Rousseau,

Please find enclosed our revised manuscript retitled "*Towards quantitative reconstruction of past monsoon precipitation based on tetraether membrane lipids in Chinese loess*". We thank Dr. David Naafs and two other anonymous reviewers for their constructive comments and your invitation to revise our work based on their feedback. Note that we have decided not to follow the suggestion by Reviewers #1 and #2 to include isoGDGT-related proxies (i.e., the BIT index and $R_{i/b}$) and the CBT(′) in the revised manuscript. Our motivation can be read in the point-to-point replies to these reviewers below. Other than that, we have followed most of their suggestions. All changes in the manuscript are made with track changes on.

To summarize, the major changes that we have made to the manuscript are:

- In the introduction, we have clarified the forcing mechanisms linked to the phasing of parameters as suggested by Reviewer #1, and further explained that both GDGTs and leaf plant waxes record in-situ signals as recommended by Reviewer #1.

- In discussion, we have described the potential link between $Ca^{2+}$ availability and changes in the soil microbial community to provide a mechanism for the DC as a precipitation proxy in loess, as requested by both Reviewer #1 and #2.

- Figures: We have changed color palette for Figure 2 (proxy panels) and Figure 4 (scatter plot). We have also modified the map (Fig. 1 and Fig. S3) as suggested by Reviewer #3. The spectral analysis of Northern Hemisphere Summer Insolation has been added to Fig. 3 as suggested by Reviewer #1.

We hope that you find this revised version suitable for publication in *Climate of the Past*.

On behalf of all co-authors,

Jingjing Guo

**Referee comments #1 (Dr. David Naafs):**

**Summary**

In this manuscript the authors reconstruct east Asian monsoon dynamics over the last 130 kyrs using biomarkers preserved in loess. Newly generated records of changes in the distribution of brGDGTs are combined with published records of plant wax d2H from the same section. The authors develop a quantitative method to reconstruct changes in MAP using brGDGTs preserved in loess and demonstrate that precipitation in the Chinese loess plateau varies at the precession and obliquity scale, the former indicating the northern hemisphere as a main driver of monsoon precipitation.

**Main assessment**

The manuscript provides a large amount of new data that are used to support novel insights into our understanding of the east Asian monsoon. This type of manuscript will be of interest to the readers of CoP. In addition, the newly proposed method to quantify precipitation using brGDGTs in loess will be of interest to organic geochemists. The manuscript was pleasant to read and the figures clear and informative.

However, my main criticism is that the manuscript and main conclusions rely on a limited set of brGDGT-based indices: DC and, to some extent, IR. However, other brGDGT indices are influenced by pH (and thus precipitation), for example the well-established CBT index for brGDGTs. In addition, other GDGTs like crenarchaeol and the BIT index can be used to infer changes in hydrology in terrestrial sections, as highlighted in the introduction of this manuscript. However, these complementary methods are not used here. Rather, the manuscript relies on the application of less often used indices like DC. There is no explanation why these other GDGT-based indices are not used, while they are measured. I

assume they are excluded because they show different results? But these other proxies could provide additional insights into changes in hydrology in this region.

I therefore recommend moderate revisions. In the revised manuscript I would like to see an expanded discussion on the other GDGT based proxies (e.g., CBT, BIT, %cren) and justification for why they are not used here to assess changes in hydrology. Or better, they are included to obtain a more holistic reconstruction of EASM dynamics across the late Quaternary.

*Reply: We thank Dr. Naafs for his positive evaluation of our work and constructive feedback. We have taken their suggestions into careful consideration and made changes in the revised manuscript accordingly. Please find our point-by-point response below in italic.*

*We have decided to not include the isoGDGT-related proxies and CBT(') proxy. A similar comment has also been made by Reviewer #2. Note that the focus on just the DC and the IR as potential proxies for monsoon precipitation is clearly motivated in the introduction of our manuscript. Specifically:*

*i) Although the BIT index and $R_{i/b}$ have been linked to hydroclimate, in loess-paleosol sequences, they are used as indicators of mega-drought events and only in a qualitative way (e.g., Xie et al., 2012; Yang et al., 2014; Tang et al., 2017). Since the aim of this manuscript is to reconstruct monsoon precipitation quantitatively, we have not included these records here. Nevertheless, the BIT index and $R_{i/b}$ in the Yuanbao section are relatively invariable and do not exceed the established threshold values (i.e., 0.5 for the $R_{i/b}$; Yang et al., 2014) that indicate the occurrence of mega-drought events at this site over the past 130 kyr (Fig. 1E).*

*ii) The environmental controls that influence the distribution of isoGDGTs in soils is still being studied, and no clear link between isoGDGTs in loess and environmental parameters*

*has been demonstrated yet. As such, the %cren does not show a clear trend in the Yuanbao*

*record (Fig. 1D). Nevertheless, the isoGDGT data will be provided as supplementary*

*material upon acceptance for the community's reference.*

*iii) The use of CBT as a proxy for monsoon precipitation has been mentioned in the*

*introduction (Line 94-102 in track-change version), as are the reasons for not using CBT*

*and/or CBT'. Namely, the first application of this proxy shows that CBT reflects monsoon*

*pacing (i.e., qualitative) rather than absolute precipitation amounts (Peterse et al., 2014).*

*Secondly, the CBT(') is in fact a combination of the degree of cyclization (DC) of brGDGTs*

*and the relative abundance of 6-methyl isomers (IR). However, the clearly opposite*

*correlations of the DC and the IR with soil pH in soils with pH >7.5 raises concerns about*

*the meaningful interpretation of CBT(') in loess sequences (e.g., Xie et al., 2012; Guo et al.,*

*2022). Hence, to determine the relationship between hydroclimate and brGDGT*

*distributions, we have deliberately split the CBT(') into the DC and the IR. As can be seen in*

*Fig. 1B, the CBT(') follows the trends in IR more than in DC, indicating that the occurrence*

*of 6-methyl brGDGTs exerts a larger influence on this proxy compared to the degree of*

*cyclisation, and the CBT('), therefore, does not align with the independent precipitation*

*indicator $\delta^2 H_{wax}$ (Fig. 1A). Therefore, we prefer to keep the focus of the manuscript on DC*

*and IR as potential proxies for monsoon precipitation, and to make both brGDGT and*

*isoGDGT data available to facilitate community efforts in improving our understanding of*

*the key parameter(s) driving their relative abundances in loess.*

[Figure]

***Fig. 1*** *Biomarker- and loess-based records for the past 130 kyr at Yuanbao. **(A)** Degree of cyclization (DC) of brGDGTs and ice-corrected $\delta^2H_{wax}$ based on plant waxes in the same lipid extracts (Fuchs et al., 2023). **(B)** Cyclization of branched tetraethers (CBT, CBT'). **(C)** Isomer Ratio (IR). **(D)** Fractional abundance of crenarchaeol to total isoGDGTs (%Cren). **(E)** Branched and Isoprenoid Tetraether (BIT) index and ratio of iso- and brGDGTs ($R_{i/b}$). **(F)** Grain size (GS) and magnetic susceptibility (MagSus). **(G)** NHSI at 35°N (Berger et al., 2010) and the composite speleothem oxygen isotope ($\delta^{18}O$) record (Cheng et al., 2016). Dark grey intervals (~23–21 ka) in brGDGT-related records (DC, IR, CBT, and CBT') indicate the transition from the outcrop to the pit and are not considered in the interpretation of the records.*

**Minor comments:**

Line 1: The method proposed here to quantify precipitation is explorative and needs to be verified at other sections. Remove "quantitative" from title to reflect the uncertainty surrounding this method.

*Reply: We thank the reviewer for their suggestion. Note that we do test the DC at two other sections for which brGDGT data are available (see section 4.2 and Fig. 5 in the original manuscript). Therefore, we feel that the DC can be considered as precipitation indicator in loess-paleosol sequences. Regardless, we have changed the title into: "**Towards** quantitative reconstruction of….".*

Line 14: state here that both the speleothem and plant wax d2H records are already published.

*Reply: We have clarified this in the revised manuscript. Line 14.*

Line 30-31: I am not an expert, but NH summer insolation also has an obliquity component, especially when we look at 65 oN and higher. In this manuscript the focus is on 35 oN insolation (e.g. figure 2), but this nuance of low versus high-latitude NH summer insolation

needs to be explained here and elsewhere in the manuscript. Also, the spectra of NH summer insolation (as shown in figure 2, so 35 oN) should be added to figure 3.

*Reply: We thank the reviewer for pointing this out. It is true that the obliquity signal in Northern Hemisphere Summer Insolation (NHSI) becomes stronger at higher latitudes. We have clarified this in the revised manuscript when discussing the precession and obliquity signals in our proxy records (e.g., Line 31). We have also added the spectrum of NHSI at 35°N to Fig. 3 in the revised manuscript.*

Line 32-34: this sentence seems to be crucial for the later interpretation of the data, but the reasoning behind this conclusion is not very clear for non-experts (like myself). The importance of this lag and why this argues against a NH insolation control needs to be explained a bit more here. This will help clarifying the discussion and conclusion later on.

*Reply: We have extended this section in the revised version of the manuscript and clarified that not only the presence of cyclicity in a proxy record but also the phasing with respect to the orbital parameters points to the forcing mechanism. See Line 33-39 in the revised manuscript.*

*Initially, Kutzbach (1981) has proposed that summer monsoon intensify with a stronger northern hemisphere summer insolation (NHSI) and would therefore vary in phase with precession cycles. Past variations in EAM climate have been inferred from proxy records such as the oxygen isotope composition ($\delta^{18}O$) of cave speleothems(Cheng et al., 2016; Wang et al., 2008, 2001) support this hypothesis (Kutzbach, 1981). However, other proxy records such as a stacked summer monsoon record from Arabian Sea shows a 6-8 kyr lag of monsoon maximum intensity with respect to the precession minima (NHSI maximum). This led to the suggestion that monsoon variations are driven by latent heat fluxes from southern*

*hemisphere as well as global ice volume (Clemens and Prell, 2003). These discrepancies*

*between proxy records opened the discussion on the interpretation of e.g. speleothem $\delta^{18}O$*

*(Clemens et al., 2010).*

Lines 46-48: Similarly, expand here to explain why a strong 23 kyr cycle is indicative for
NHSI.

*Reply: We have added that precession represents a 23-kyr cycle as is also present in NHSI .*
*See e.g., Line 31 in the revised manuscript.*

Line 68-onwards: somewhere in this section of the introduction of the manuscript explain
where (and when) the biomarkers that are found in loess are produced. Do this for both the
GDGTs and the plant waxes. For example, are the plant waxes produced in situ or transported
with the loess? This nuance is important for the later discussion.

*Reply: We thank the reviewer for raising this important point. In the loess-biomarker*

*literature, both GDGTs and n-alkanes are commonly assumed to reflect an in situ signal.*

*This is based on the absence of GDGTs in material from loess source regions, mainly due to*

*the unfavorable growth conditions for GDGT producers in these arid deserts (Gao et al.*

*2012). In addition, GDGTs were below detection limit in wind-transported dust, suggesting*

*that they are not commonly transported through the atmosphere (Hopmans et al., 2004).*

*Similarly, the plant wax signals can be interpreted as a local signal. Specifically for*

*Yuanbao, the carbon isotope signal of the plant waxes indicates a consistent dominance of $C_3$*

*vegetation, which aligns with the high elevation and cold, dry winters at Yuanbao that are*

*unfavorable for $C_4$ plants (Fuchs et al., 2023). Secondly, vegetation is sparse in the main dust*

*source for the CLP, located northwest of Yuanbao. During summer, when the main wind direction is east-to-west, dust-associated transport of biomarkers is unlikely due to the higher vegetation cover and increased precipitation towards the east, which largely prevents dust mobilization. These processes have also been discussed and confirmed by previous studies (Liu et al., 2005; Thomas et al., 2016; Zhou et al., 2016).*

*We have specified this in the revised manuscript. Line 108-110.*

Line 83: this is a bit of a NIOZ/UU centered list of papers. Lots of other groups have worked on this, I suggest diversifying the reference list here.

*Reply: We thank the reviewer for their comments, we have added other works that highlight the influence of growing season temperature on brGDGT production to the revised manuscript. Line 88-89.*

Line 86-89: I was surprised that CBT was not discussed at all here (and not used at all in the entire manuscript). CBT is one of the most common methods to reconstruct soil pH. It needs to be introduced here. In this context, I wonder whether changes in the accumulation rates of brGDGTs hold any paleoclimatic information. The GDGTs were quantified using the C46 std, so this data is available.

*Reply: We thank the reviewer for pointing this out. The CBT and CBT' are definitely on our checklist, as always. However, as we mentioned in our response to your earlier comments, and also explain in the introduction of our manuscript, both CBT and CBT' are combinations of the degree of cyclization (DC) and the isomer ratio (IR) (Fig. 1B). In arid and alkaline soils, however, it has been found that the DC and the IR exhibit opposite correlations with soil pH > 7 (Guo et al., 2022; and Fig. 4A and B in this manuscript). This finding is in*

*accordance with the previously observed abrupt change in the relationship between CBT and soil pH at pH = 7.5 (Xie et al., 2012). Furthermore, the abundance of brGDGT compounds shows different optimal pH ranges (e.g., Supp Fig. 3 in De Jonge et al., 2021). Hence, the environmental parameter(s) that control the CBT in alkaline soils (like loess) deviates from the global trend and is not well understood. As we also explain in the Introduction, we have, therefore, decided to focus on DC and IR. With this approach, we aim to improve our understanding of the key factors influencing all different aspects of changes in brGDGT distributions in loess.*

*As for the GDGT concentrations, these data are indeed available. As shown in Fig. 2A and B, GDGT concentrations follow the same trend as magnetic susceptibility (MagSus, Fig. 2D), indicating that they are similarly impacted by sedimentation rates (dilution) and/or the rate of soil formation (production) as MagSus. As such, this record does not provide additional paleoclimatic information beyond what MagSus already indicates. We have therefore decided not to include the CBT(') and concentration records to the revised manuscript.*

[Figure]

***Fig. 2*** *Biomarker- and loess-based records for the past 130 kyr at Yuanbao. **(A)** Concentration of isoprenoid GDGTs. **(B)** Concentration of brGDGTs. **(C)** Grain size (GS) and magnetic susceptibility (MagSus). **(D)** NHSI at 35°N (Berger et al., 2010) and the composite speleothem oxygen isotope ($\delta^{18}O$) record (Cheng et al., 2016).*

Line 88: methylation can also occur at C7, see for example (Ding et al., 2016)

*Reply: We have added this to the revised manuscript. Line 95-96.*

Line 90: we also discussed this in (Naafs et al., 2017)

*Reply: We have included this in the revised manuscript. Line 95.*

Line 113: and is this benthic d18O record tuned to astronomical cycles like the LR04 stack is?

*Reply: This benthic $\delta^{18}O$ record used for our age model is LR04, we have specified this in the revised manuscript to avoid any confusion. Line 122.*

Line 116: change to "…corresponding to a sedimentation…"

*Reply: We have changed this in the revised manuscript. Line 126.*

Line 144: Explain here why IIc and IIIb-IIIc are not used in the DC index. Is their abundance too low?

*Reply: These compounds are indeed mostly below detection limit in the Yuanbao sequence. In addition, the equation for the DC presented in Baxter et al. (2019) does not include these compounds, see Eq. 2 in their paper. Regardless, including the (low contributions of) brGDGT-IIc('), -IIIb(') and -IIIc(') in the DC does not affect the overall trend of the DC record for Yuanbao (Fig. 3), and the offset between DC with and without hexamethylated brGDGTs are mainly induced by brGDGT-IIIa. For the DC-MAP calibration, using the same*

*DC equation for both calibration and downcore records calculation will not impact the absolute precipitation reconstructions.*

*DC = ([Ib]+2\*[Ic]+[IIb]+[IIb'])/([Ia]+[Ib]+[Ic]+[IIa]+[IIa']+[IIb]+[IIb'])*

*DC (with hexamethylated brGDGTs) =*

*([Ib]+2\*[Ic]+[IIb]+[IIb']+2\*[IIc]+2\*[IIc']+[IIIb]+[IIIb']+2\*[IIIc]+2\*[IIIc'])/([Ia]+[Ib]*
*+[Ic]+[IIa]+[IIa']+[IIb]+[IIb']+[IIc]+[IIc']+[IIIa]+[IIIa']+[IIIb]+[IIb']+[IIIc]+ IIIc'])*

[Figure]

**Fig. 3** *Degree of Cyclization (DC) over the past 130 kyr at Yuanbao using the original equation (orange curve, Baxter et al., 2019) and including the hexamethylated brGDGTs (purple curve). Dark grey intervals (~23–21 ka) in brGDGT-related records (DC, IR, CBT, and CBT') indicate the transition from the outcrop to the pit and are not considered in the interpretation of the records.*

Line 153: what is this assumed standard deviation based on? Can repeat analysis of for example a lab standard provide a data-supported value? If not, what is the impact of selecting a slightly different value? How does this impact the MAP reconstructions?

*Reply: The standard deviation of the DC in our lab is 0.02 based on an in-house standard run every ~10 samples. For this study, we chose 0.05 to ensure that potential between-lab variations are better accounted for in the calibration. Nevertheless, using a standard deviation of 0.02 or 0.05 does not significantly impact the estimated MAP, as both fall within the calibration uncertainty (± 125 mm). Specifically, assuming a given DC of 0.5, a standard deviation of 0.02 yields a MAP of 784 mm, while a standard deviation of 0.05 yields a MAP of 804 mm.*

Line 177-179: show cross plots of DC versus GS and d2H to provide a statistical basis for this "match"

*Reply: The cross plots of DC vs GS and DC vs $\delta^2 H_{wax}$ are shown in Fig. 4. We have decided not to add them into the manuscript because: i) It is evident in the time series that the low DC corresponds with more negative $\delta^2 H_{wax}$ and high GS, and that the timing and direction (but not necessarily the amplitude) of changes in these records is similar. ii) The main focus in the discussion here is on the Henrich stadials (HS; i.e., millennial-scale events) during the last glacial period (Fig. 4C and D), rather than the entire record (Fig. 4A and B). As we have discussed in the manuscript (e.g., Line 285-293 in the original manuscript), the evapotranspiration and changes in moisture source impact the trend of $\delta^2 H_{wax}$ which is not related to the precipitation amount recorded by DC. Regardless, the HS are clearly present in GS, DC and $\delta^2 H_{wax}$.*

[Figure]

**Fig. 4** *Cross plot of (**A**) DC vs GS over the past 130 kyr and (**C**) the last glacial period; (**B**) DC vs $\delta^2 H_{wax}$ over the past 130 kyr and (**D**) the last glacial period. The color gradients indicate age of downcore samples.*

Line 179: also show cross plot for NH insolation and the IR record for comparison (and for other proxies used, see main comment)

*Reply: We have decided not to include the cross plots of NHSI and GDGT-based proxies because DC and other GDGT-based proxies record sub-Milankovitch and millennial-scale variability. NHSI is influenced by orbital cycles and contains no sub-Milankovitch variability. For this reason, we included the bandpass filters of DC (Fig. 2B in the original manuscript) to directly compare with NHSI and highlight the signal of orbital forcings in our DC record.*

Line 182: this grey is hard to see, add arrow to figure of where this splicing occurs

*Reply: We have adjusted this in the revised manuscript.*

Figure 3: How do you get 100 kyr cyclicity in a 130 kyr long record?

*Reply: We agree that discussing glacial-interglacial cycles with a record that only extends back 130 kyr is challenging. Therefore, we consistently state that the interpretation of glacial-interglacial cycles in our record is restricted by the length of our record (e.g., Line 299 in the original manuscript). Nevertheless, the presence of a 100 kyr cycle in loess proxy records (GS and MagSus) is well described in earlier work based on longer time series (e.g., Sun et al., 2022).*

Line 195: but besides precession, the DC record also shows a strong 41 kyr signal

*Reply: Indeed, both precession and obliquity signals are evident in the DC record (Fig. 3A in the original manuscript). However, in this part of the manuscript, our primary focus is the*

*comparison between DC and $\delta^2 H_{wax}$, we have therefore mainly discussed their shared*

*characteristics, particularly those in the precession band. The discussion of obliquity cycles*

*occurs later in the manuscript, i.e., Line 315.*

Line 199: to fully determine whether the DC and IR records are different, show a spectral

analysis of the IR record to highlight that it lack precession and obliquity cycles as seen in the

DC record.

*Reply: The interpretation of IR in Yuanbao section is not clear (yet), therefore we have not*

*included any timeseries analysis of this record in the previous version, but for your*

*information, the spectral analysis of IR is shown below (Fig. 5F). Although the IR record*

*does contain a precession signal, its spectral density is comparatively smaller than that of the*

*other proxies (Fig. 5F). Taken together, we have therefore decided not to add the spectral*

*density of the IR record into the revised manuscript.*

[Figure]

**Fig. 5** *Spectra of time series of proxy records from the Yuanbao loess section, cave speleothems in southeast China, and Northern Hemisphere Summer Insolation (NHSI) at 35°N.*

Line 207: could the brGDGTs be used to quantify these variations in hot and cold conditions?

*Reply: Unfortunately, no. The MBT'$_{5ME}$ index shows an abrupt and large increase in parallel with the change in IR. Next to the fact that this would suggest that conditions changed from cold to warm, which is opposite to what is suggested by the loess proxies and isotopic signals based on speleothem δ$^{18}$O and δ$^2$H$_{wax}$, the MBT'$_{5ME}$ also shows non-analogue behavior in part of the core. This introduces uncertainty in the brGDGT-temperature relationship. The MBT'$_{5ME}$ record for this section and an extensive assessment of its environmental controls is part of a manuscript that is currently under review with Organic Geochemistry.*

Line 211: doesn't Ca2+ affect pH and that influences brGDGT production? Is there clear evidence that it is Ca2+ and not the resulting change in pH?

*Reply: We agree with the reviewer that distinguishing the impacts of $Ca^{2+}$ and soil pH on brGDGT distributions is challenging. Nevertheless, loess is generally rich in carbonate, which will dissolve after rainfall. The then released $Ca^{2+}$ could possibly contribute to an increase in soil pH. However, this mechanism points at the amount of available $Ca^{2+}$ as the primary driver of the production of cyclic brGDGTs in loess. In addition, available $Ca^{2+}$ was found to be a more important factor explaining the relative abundance of cyclic brGDGTs in an Arctic elevation transect (Halffman et al., 2022) as well as a suite of mid-latitude soils (De Jonge et al., 2021) compared to soil pH or free acidity.*

Line 215: I don't understand how Ca2+ affects brGDGT production. The direct impact of Ca2+ on brGDGT producers needs explanation. If Ca2+ drives pH and that impacts brGDGT producers, explain that here

*Reply: Unfortunately, the exact link between available $Ca^{2+}$ and the production of cyclic brGDGTs is currently based on empirical correlations in soils from Scandinavia (Halffman et al., 2022) as well on more global soil datasets (De Jonge et al., 2021, 2024). However, microbial ecological studies have shown that Ca rather than pH is a key predictor in shaping the soil microbiome as well as its functionality (e.g., Shepherd and Oliverio, 2024; Neal and Glendining, 2019; Allison et al., 2007). We have added the information on the influence of Ca on the microbial diversity in soils to the revised manuscript. Line 228-230.*

Line 222: doesn't a r2 of 0.06 indicate no correlation?

*Reply: We have rephrased this in the revised manuscript. Line 240.*

Line 225: although the overall community might change, this doesn't mean that the brGDGT producing community changes. The next sentence should state that this is speculation.

*Reply*: *We have rephrased this in the revised manuscript. Line 243-244.*

Figure 4: do I understand correction from this figure that pH is not correlated to MAP because IR is strongly correlated with pH, but not with MAP? Does that not undermine some of the earlier text of this manuscript where MAP and pH are suggested to be linked?

*Reply*: *As the figure below shows, MAP and soil pH stills show a negative correlation globally (Fig. 6). In general, IR also relates to MAP if we do not separate the data into different groups based on soil pH.*

[Figure]

**Fig. 6** *Cross plots of observed mean annual precipitation (MAP) vs soil pH and isomer ratio.*

Line 243: cite reference for modern soil CLP GDGT data here

*Reply: We have included the reference in the revised manuscript. Line 262.*

Line 245: how does this uncertainty of ±125 mm compare to other quantitative proxies used for the CLP? Is this correlation much better, worse, or similar to other methods? This context would be good for the non-expert.

*Reply: The uncertainty of quantitative precipitation reconstructed by $^{10}$Be is 190 mm (1 standard deviation) (Beck et al., 2018), we have specified this in the revised manuscript. Line 286.*

*The uncertainty of microcodium Sr/Ca ratio is not reported (Li et al., 2017).*

Line 266: and how does this gradient compare to the modern gradient?

*Reply: The modern MAP at Yuanbao, Xifeng and Weinan is 500 mm, 470 mm and 600 mm, respectively. The downcore records reflect a similar spatial gradient as seen in the modern observations. We have clarified this in the revised manuscript. Line 291.*

Line 268: is the difference in reconstructed MAP between Holocene optimum and MIS 5e statistically not different? State statistically proof for this statement.

*Reply: In this sentence we aimed at making the point that reconstructed MAP was spatially similar during interglacials, i.e., between sites, but not between the Holocene and MIS5, as our data indeed suggest that MAP was higher during MIS5 than during the Holocene. We have rephrased this sentence for clarification in the revised manuscript. Line 289.*

Line 283: the manuscript states a "close resemblance", but the DC has a strong 41 kyr signal that is lacking in the d2H record.

*Reply: We thank the reviewer for pointing this out, we have rephrased this in the revised manuscript to ensure that the description is more precise. Line 305-306.*

Lines 283-308: For this comparison with the d2H record, it is important to in the introduction explain where and when the different biomarkers are produced. Is there a possibility for a spatial and/or temporal offset between production of the plant waxes and bacterial membrane lipids?

*Reply: As explained in our response to your earlier comments, both the brGDGT-based proxies and $\delta^2H_{wax}$ at Yuanbao reflect in-situ signals and this should not have spatial or temporal offsets. We have clarified this in the Introduction in revised manuscript. Line 108-110.*

Figure 6: why is the IR data not included here? Does that not have a clear precession forcing?

*Reply: The main reason we decided not to include IR in the phase wheel is that it remains unclear which climate aspect IR indicates in this downcore record, while the phase wheel is intended to show the leads and lags of precipitation-related proxies on the precession band. In addition, as the reviewer has pointed out, although the IR shows a precession signal based on the spectral analysis, its spectral density is comparatively smaller than that of the other proxies (Fig. 5)*

Line 343: ensure that the individual GDGT data is available for future usage

*Reply: We thank the reviewer for their reminder. The related dataset has been submitted to PANGAEA database. We have added the DOI link to the revised manuscript. In addition, we have deposited our dataset on Open Science Frame and a link has been added to the revised manuscript. Line 365.*


**Referee comment #2:**

Guo and co-authors present a novel dataset, and apply the DC index as a proxy for precipitation, in alkaline soils from a loess sequence. The paper is in general very well written, the figures are clear and the paper is well structured. The authors did a very good job explaining the result, that at first sight seem to contradict the expected GDGT responses. I do have a few comments that could broaden the interest of the manuscript to users of GDGT proxy ratios.

*Reply: We thank the reviewer for their positive evaluation of our work. We have taken your suggestions into careful consideration and will make clarifications accordingly in the revised manuscript. Please find our point-by-point response below in italic.*

The discussion of the results is very to-the-point, to a degree where I wonder whether more GDGT ratios (Ri/b and BIT seem obvious choices (see introduction), possibly together with GDGT concentrations or accumulation rates) could have been included.

*Reply: A similar comment has also been made by Referee #1. Note that the focus on just the DC and the IR as potential proxies for monsoon precipitation is clearly motivated in the introduction of our manuscript. Specifically, although the BIT index and $R_{i/b}$ have been linked to hydroclimate, in loess-paleosol sequences, they are used as indicators of mega-drought events and only in a qualitative way (e.g., Xie et al., 2012; Yang et al., 2014; Tang et al., 2017). Since the aim of this manuscript is to reconstruct monsoon precipitation quantitatively, we have decided not to include these records in the revised manuscript. Nevertheless, the BIT index and $R_{i/b}$ in the Yuanbao section are relatively invariable and do not exceed the established threshold values (i.e., 0.5 for the Ri/b; Tang et al., 2017) that indicate the occurrence of mega-drought events at this site over the past 130 kyr (Fig. 1B).*

[Figure]

***Fig. 1*** *Biomarker- and loess-based records for the past 130 kyr at Yuanbao. **(A)** Degree of cyclization (DC) of brGDGTs and ice-corrected $\delta^2 H_{wax}$ based on plant waxes in the same lipid extracts (Fuchs et al., 2023). **(B)** Branched and Isoprenoid Tetraether (BIT) index and ratio of iso- and brGDGTs ($R_{i/b}$). **(C)** Grain size (GS) and magnetic susceptibility (MagSus). **(D)** NHSI at 35°N (Berger et al., 2010) and the composite speleothem oxygen isotope ($\delta^{18}O$) record (Cheng et al., 2016). Dark grey intervals (~23–21 ka) in brGDGT-related records (DC) indicate the transition from the outcrop to the pit and are not considered in the interpretation of the records.*

*The GDGT concentrations follow the same trend as magnetic susceptibility (MagSus, Fig. 2), indicating that they are similarly impacted by sedimentation rates (dilution) and/or the rate of soil formation (production) as MagSus. As such, this record does not provide additional paleoclimatic information beyond what MagSus already indicates. We have therefore decided not to include them in the revised manuscript.*

[Figure]

**Fig. 2** *Biomarker- and loess-based records for the past 130 kyr at Yuanbao.* **(A)** *Concentration of isoprenoid GDGTs.* **(B)** *Concentration of brGDGTs.* **(C)** *Calcium (Ca) intensity measured by X-ray fluorescence (unit: counts per second, cps) from a nearby drilling core (Guo et al., 2021).* **(D)** *Grain size (GS) and magnetic susceptibility (MagSus).* **(E)** *NHSI at 35°N (Berger et al., 2010) and the composite speleothem oxygen isotope ($\delta^{18}O$) record (Cheng et al., 2016).*

I would invite the authors to think a bit further about the local paleo-environmental conditions, and whether any of their assumptions can be confirmed with independent measurements. For instance, the authors surmise a link with a change in DC and available Ca in the soil profile. Could this be substantiated by the analysis of the sedimentology of this sequence?

*Reply: We thank the reviewer for their suggestions. We have checked Ca variations in a nearby drilling core at Yuanbao (103.63°E, 35.15°N, 2200 m above sea level; Guo et al., 2021), which was conducted by X-ray fluorescence (XRF) scanning (Fig. 2C).*

*As discussed in the manuscript, more cyclic brGDGTs are produced when $Ca^{2+}$ becomes increasingly available under wet conditions due to the dissolution of $CaCO_3$ that is generally present in loess (Line 209-217 in the original manuscript). Indeed, the DC and Ca intensity show a similar trend over the past 130 kyr (Fig. 2A and C).*

*Although the exact link between available $Ca^{2+}$ and the production of cyclic brGDGTs is currently based on empirical correlations in soils from Scandinavia (Halffman et al., 2022) as well on more global soil datasets (De Jonge et al., 2021, 2024), microbial ecological studies have shown that Ca is a key predictor in shaping the soil microbiome as well as its functionality (e.g., Shepherd and Oliverio, 2024; Neal and Glendining, 2019; Allison et al., 2007). We have added the information on the influence of Ca on the microbial diversity in soils to the revised manuscript. Line 228-230.*

I also wonder whether the short-lived hydrological shift that resulted in a long impact on the 6-methyl branched GDGTs, could have been caused by ponding (i.e. the creation of a lake)? Is there any evidence for stagnant water (and associated anoxia) based on the sediments?

*Reply: We thank the reviewer for their thoughtful brainstorming. It is indeed an interesting point to consider whether stagnant water might have occurred during the period when the IR shifts. However, this does not appear to be the case at Yuanbao section.*

*First, we do not expect stagnant water in loess-paleosol sequences, let alone anoxic conditions. While heavy precipitation may lead to temporary saturation of the topsoil, excess water would likely rapidly evaporate due to the generally warm climate conditions during the monsoon season, penetrate deeper into the soil due the loose texture of loess, or runoff as a result of relief on the CLP. In addition, the shift in 6-methyl brGDGTs occurs during a glacial period, which, on the CLP, is characterized by arid conditions. Moreover, the ratio of GDGT-0/crenarchaeol, which indicates contributions of methanogens when values exceed 2 (Blaga et al., 2009), is 0.1–1.7 in the Yuanbao section. Also the grain size record does not show any indications for a change in depositional environment. Taken together, we exclude the presence of stagnant waters at this location during this time interval.*

The different signal between pit and outcrop is also interesting, and points to the large impact local (hydroclimate/ vegetation) changes have on the GDGT signal.

*Reply: We agree with the reviewer that the large shift in brGDGT signals during the transition from the outcrop to the pit is intriguing. However, the other proxies from the same section (i.e., traditional loess proxies, $\delta^2H_{wax}$ from leaf waxes and isoGDGTs) do not show this difference, therefore it may be an overinterpretation to draw conclusions based solely on*

*the shift in brGDGTs. Nevertheless, we have clarified that the transition between the pit and*

*outcrop is not considered in our discussion (Line 195-198).*

*References*

[revised manuscript text omitted]

**Referee comment #3:**

Guo et al. generated new records based on branched glycerol dialkyl glycerol tetraethers (brGDGTs) from the Chinese Loess Plateau (CLP) over the last 130 kyrs. The authors found that two pH-sensitive brGDGT-based indices, DC and IR, showed contrasting temporal changes at the same site. After a comparison of the new brGDGT-based records with several published records from the same site, such as another biomarker-based record (ice-corrected δ2Hwax), the authors found that DC showed promise as a mean annual precipitation (MAP) proxy. Then, the authors investigated the relationships between brGDGT-based indices (DC and IR) and soil pH and MAP using a global modern soil dataset, as well as a CLP modern soil dataset. The authors found that in alkaline soils, including within the CLP, DC showed a strong correlation with MAP, which enabled the development of a DC-MAP calibration for quantitative MAP reconstructions. The authors then applied their DC-MAP calibration at three sites within the CLP, including the study site, and investigated spatial differences in MAP within the CLP and their changes through time. The authors also did spectral and cross-spectral analyses and found that (i) their DC record showed precession and obliquity signals, contrary to δ2Hwax and δ18Ospeleorecords which only showed the precession signal, and (ii) their DC record was in phase with δ2Hwax and δ18Ospeleorecords at the precession scale. The authors thus concluded that Northern Hemisphere summer insolation was a direct forcing of precipitation amount rather than the result from confounding factors on precipitation records based on isotopes.

As a paleoclimatologist with expertise on GDGT-based proxies, I have read this manuscript with interest for several reasons. First, the authors tackled a topical and important subject, namely the understanding of the East Asian Monsoon which also has its own controversies and subjects of debate, as the authors stated in the Introduction. Second, this manuscript presents an interesting use of a brGDGT-based proxy, namely DC as a (quantitative) MAP

proxy, as brGDGTs are classically used for quantitative reconstructions of land temperature and soil pH. Third, the authors reconstructed past MAP changes using a proxy that does not involve hydrogen or oxygen isotopes, contrary to $\delta^2H_{wax}$ and $\delta^{18}O_{speleo}$, which strengthens the independence of DC relative to $\delta^2H_{wax}$ and $\delta^{18}O_{speleo}$. Furthermore, I found the manuscript easy to read and well-organized. Overall, this piece of work would be a great addition to the literature and is worth publishing in Climate of the Past.

However, I have several comments, questions, and suggestions for revision, which I detail below.

*Reply: We thank the reviewer for their positive evaluation of our work. Please find our point-by-point response below in italic.*

General comments:

1) Recently, Zhao et al. (2020) introduced a precipitation index (PI) as follows: PI = (Ia + Ib)/(Ia + Ib + IIIa + IIa′ + IIIa′). Like the redefined DC, the PI takes advantage of the improved separation of 5- and 6-methyl brGDGT isomers. Importantly, Zhao et al. (2020) and Zhang et al. (2024) proposed PI-MAP calibrations in cancellous bones and soils, respectively, for brGDGT-based quantitative MAP reconstructions. Accordingly, I would like to know the authors' thoughts concerning the PI. Specifically, the authors may check how well the PI would behave as a (quantitative) MAP proxy compared with e.g., DC and IR in the CLP and, in case of similar trends between DC and PI, which MAP reconstructions the PI would yield in the CLP. However, the authors do not need to switch to the PI, especially if DC has a better motivation and/or shows more meaningful results compared with the PI in the authors' view.

Zhang, T., Han, W., Tian, Q., Zhang, J., Kemp, D. B., Wang, Z., Yan, X., Mai, L., Fang, X., and Ogg, J.: Tectonically controlled establishment of modern-like precipitation patterns in East and central Asia during the early late Miocene, Journal of Geophysical Research: Atmospheres, 129, e2024JD041025, https://doi.org/10.1029/2024JD041025, 2024.

Zhao, J., Huang, Y., Yao, Y., An, Z., Zhu, Y., Lu, H., and Wang, Z.: Calibrating branched GDGTs in bones to temperature and precipitation: application to Alaska chronological sequences, Quaternary Science Reviews, 240, 106371, https://doi.org/10.1016/j.quascirev.2020.106371, 2020.

*Reply: We thank the reviewer for their comments and providing the references. In previous studies, the precipitation index (PI) has shown a positive correlation with precipitation in bones and surface soils from China (Zhao et al., 2020; Zhang et al., 2024). The application of the PI in the Yuanbao section, however, results in a record with higher values during stadials than interstadials during MIS5, which is inconsistent with other hydroclimate proxies from the same section (Fig. 1B), such as MagSus and $\delta^2H_{wax}$. In addition, compared to the DC, the PI fails to capture millennial-scale events during the last glacial period.*

*Upon closer examination, the PI equation combines the degree of methylation (MBT), the position of the methylation (IR), and the degree of cyclization (DC), making it complex to interpret which membrane adjustment is driving the changes in the PI record. Specifically, having the tetramethylated brGDGTs in the numerator and the penta- and hexamethylated brGDGTs in denominator resembles the structure of the MBT'$_{5Me}$, which is linked to temperature -and not precipitation- in the global soil dataset (De Jonge et al., 2014). In addition, the PI was initially developed based on distributions of brGDGTs in bones, where bacterial community compositions might differ from those in (arid and alkaline) soils and loess and paleosol materials.*

*For these reasons, we have decided not to include the PI in our revised manuscript.*

*Nevertheless, it would be interesting to test the PI in different types of material in future*

*work.*

[Figure]

**Fig. 1** *Biomarker-based records for the past 130 kyr at Yuanbao. (A) Degree of cyclization (DC) of brGDGTs and ice-corrected δ²H_{wax} based on plant waxes in the same lipid extracts (Fuchs et al., 2023). (B) brGDGT-based precipitation index (PI) (Zhao et al., 2020). (C)*

*isoGDGT-based mean monthly precipitation (MMP) record (De Jonge et al., 2024). (**D**) NHSI at 35°N (Berger et al., 2010) and the composite speleothem oxygen isotope ($\delta^{18}O$) record (Cheng et al., 2016). Dark grey intervals (~23–21 ka) in brGDGT-related records (DC and PI) indicate the transition from the outcrop to the pit and are not considered in the interpretation of the records.*

2) Even more recently, De Jonge et al. (2024a) proposed another GDGT-based proxy which may track precipitation changes, specifically mean monthly precipitation (MMP) changes: MMP = (isoGDGT-1 + isoGDGT-3)/(isoGDGT-1 + cren). I recognize that the involved GDGTs are isoGDGTs rather than brGDGTs and that the alternative GDGT-based index would likely yield uncertain reconstructions as well (see the [Eq. 14] versus MMP plot in Supp. Fig. 8 in De Jonge et al., 2024a). Nevertheless, provided that isoGDGTs are abundant enough to yield peak areas above quantification limit, I feel that the authors may consider this isoGDGT-based ratio as well.

De Jonge, C., Guo, J., Hällberg, P., Griepentrog, M., Rifai, H., Richter, A., Ramirez, E., Zhang, X., Smittenberg, R. H., Peterse, F., Boeckx, P., and Dercon, G.: The impact of soil chemistry, moisture and temperature on branched and isoprenoid GDGTs in soils: a study using six globally distributed elevation transects, Organic Geochemistry, 187, 104706, https://doi.org/10.1016/j.orggeochem.2023.104706, 2024a.

*Reply: We thank the reviewer for highlighting this isoGDGT-based precipitation proxy. We have applied the proxy to our loess section (Fig. 1C). However, the resulting record does not show a trend that could be linked to any climate events, regionally or globally. Although the work of De Jonge et al., (2024) suggests that isoGDGTs in soils have potential as a proxy for precipitation reconstructions, the drivers of isoGDGT distributions in arid and alkaline soils, like loess, remain elusive. At least, the mean monthly precipitation (MMP) record we obtained for Yuanbao suggests that precipitation may not be the dominant control. We have therefore decided not to include this record into our revised manuscript.*

Detailed comments:

Main text

Line 11: In "Chinese Loess Plateau", "loess" is not capitalized in the abstract, but is in line 40 in the Introduction.

*Reply: We have corrected this in the revised manuscript. Line 11.*

Line 48: Which paper by Guo et al. (2022) is cited here? The one published in Geology (Guo et al., 2022a)?

*Reply: It is the Geology one indeed. We have made this clear in the revised manuscript. Line 53.*

Line 98: Which paper by Guo et al. (2022) is cited here? The one published in Organic Geochemistry (Guo et al., 2022b)?

*Reply: We thank the reviewer for their careful check, this is the one published in Organic Geochemistry. We have made it clear in the revised manuscript. Line 104.*

Fig. 1 (also Fig. S3): Readers may find it hard to read the coordinate labels for the inset which shows the relevant wind patterns: the authors should consider changing the color from black to white, as they did for other labels within the larger map, and/or increasing the font size.

*Reply: We have modified this accordingly in the revised manuscript.*

Lines 142–143: Replace "(De Jonge et al., 2014a)" with "De Jonge et al. (2014a)".

*Reply: We have replaced this in the revised manuscript. Line 155-156.*

Fig. 2: For panel A, the authors should pick a color pair different from the current green-orange one for the sake of accessibility to color-blind readers. For panel B, the authors could consider picking a color pair with a stronger contrast in terms of hue and/or lightness for the sake of readability.

*Reply: We have changed this in the revised manuscript.*

Line 204: Replace "this event only last" with "this event only lasts".

*Reply: We have corrected this in the revised manuscript. Line 220.*

Fig. 4: For panels A–D, the authors should consider revising the colors to avoid the green-orange confusion for color-blind readers. Alternatively, the authors should distinguish the CLP datapoints from the other ones using a different symbol type, for instance with squares, diamonds, or triangles rather than circles. If the authors pick the second option, then the change in symbol type for CLP should be reflected in panels E and F as well.

*Reply: We have modified this accordingly in the revised manuscript.*

Lines 247–248: The r2 value represents the percentage of variance explained by the regression, not the correlation strength which is represented by the r value.

*Reply: We have changed this in the revised manuscript (e.g., Figure 4, Line 239).*

Lines 255–258: This is an important and welcome remark.

*Reply: We thank the reviewer for their positive feedback.*

Line 294: "(i.e., δ2Hwax (Fuchs et al., 2023), speleothem δ18O (Cheng et al., 2016))": A few parentheses should be removed so that only a pair of parentheses remains.

*Reply: We have carefully checked and changed in the revised manuscript. Line 316.*

References: Could the authors recheck their reference list? The formatting appears a bit suboptimal at places, for instance in lines 367–368 (Baxter et al., 2019) where I spotted a "ScienceDirect" which appears out of place there, as well as in lines 537–539 (Wang et al., 2001) where I spotted an unexpected "(80-. )." just after the journal name.

*Reply: We thank the reviewer for their careful check. We have carefully went through the reference list and corrected the formatting.*

Supplementary Figures

Fig. S1: It would be great if the authors could write the m/z values of [M+H]+ ions with at least one decimal place rather than as integer values. Otherwise, researchers who would examine GDGTs for the first time may fail to do optimal GDGT analyses for the reasons discussed by Davtian et al. (2018) and partly reminded by De Jonge et al. (2024b).

Davtian, N., Bard, E., Ménot, G., and Fagault, Y.: The importance of mass accuracy in selected ion monitoring analysis of branched and isoprenoid tetraethers, Organic Geochemistry, 118, 58–62, https://doi.org/10.1016/j.orggeochem.2018.01.007, 2018.

De Jonge, C., Peterse, F., Nierop, K. G. J., Blattmann, T. M., Alexandre, M., Ansanay-Alex, S., Austin, T., Babin, M., Bard, E., Bauersachs, T., Blewett, J., Boehman, B., Castañeda, I. S., Chen, J., Conti, M. L. G., Contreras, S., Cordes, J., Davtian, N., van Dongen, B., Duncan, B., Elling, F. J., Galy, V., Gao, S., Hefter, J., Hinrichs, K.-U., Helling, M. R., Hoorweg, M., Hopmans, E., Hou, J., Huang, Y., Huguet, A., Jia, G., Karger, C., Keely, B. J., Kusch, S., Li, H., Liang, J., Lipp, J. S., Liu, W., Lu, H., Mangelsdorf, K., Manners, H., Martinez Garcia, A., Menot, G., Mollenhauer, G., Naafs, B. D. A., Naeher, S., O'Connor, L. K., Pearce, E. M., Pearson, A., Rao, Z., Rodrigo-Gámiz, M., Rosendahl, C., Rostek, F., Bao, R., Sanyal, P., Schubotz, F., Scott, W., Sen, R., Sluijs, A., Smittenberg, R., Stefanescu, I., Sun, J., Sutton, P., Tierney, J., Tejos, E., Villanueva, J., Wang, H., Werne, J., Yamamoto, M., Yang, H., and Zhou, A.: Interlaboratory comparison of branched GDGT temperature and pH proxies using soils and lipid extracts, Geochemistry, Geophysics, Geosystems, 25, e2024GC011583, https://doi.org/10.1029/2024GC011583, 2024b.

*Reply: We thank the reviewer for their careful review and detailed suggestions. We have added this important information to the Method and supplementary figure in the revised manuscript to avoid any potential misleading. Line 146-148.*

---

## Author Response (AR2)

Dear authors,

I would like to thank you for substantially revising your initial manuscript, taking into account most of the reviewers' comments and suggestions. I agree with them that you have done an excellent job with your revision. However, there are still some minor points that the reviewers, at least two of them, would like you to improve, asking to check this final revision. This is why I have selected "major revision", as this is the only option that the Copernicus system allows me to involve these two reviewers without going back on the scientific merit of your revision. Please take these final adjustments into account in this latest revision.

I look forward to reading from you.

All the very best

denis-didier Rousseau

CP co-editor in chief

Dear editor, dear Prof. Rousseau,

Thank you for your positive evaluation of our revision. Please find enclosed our revised manuscript titled "*Towards quantitative reconstruction of past monsoon precipitation based on tetraether membrane lipids in Chinese loess*". We thank Dr. David Naafs and two other anonymous reviewers for their second-round comments and your invitation to revise our work based on their feedback. We have followed most of their suggestions and have included the concentration of iso- and brGDGTs to Fig. 2 in the manuscript as suggested by Reviewer #1, and the CBT′ and iso-GDGT related proxies (i.e., $R_{i/b}$ and BIT) to the supplementary figure as suggested by Reviewer #1 and #2. All changes in the manuscript are made with track changes on.

We hope that you find this revised version suitable for publication in *Climate of the Past*.

On behalf of all co-authors,

Jingjing Guo

**Referee comments #1 (Dr. David Naafs):**

**Main assessment**

-I like the new title

*Reply*: *We thank Dr. Naafs for his positive evaluation of the new title.*

-The isoGDGT and CBT' and other indices as shown in figure 1 of the reply letter should be included in the main manuscript and not hidden in the supplementary information. Exploring this data is informative and provides further insights into the use of biomarkers in loess.

*Reply*: *We thank Dr. Naafs for his comment. However, as we explained in our previous reply, CBT' is a proxy that includes changes in both the degree of cyclization and isomerization (see Eq. 1), and therefore does not allow us to identify which adaptations in the molecular structure of the brGDGTs is driving changes in the CBT'. From our Fig. 2 in the manuscript it is evident that DC and IR show different trends in the Yuanbao section, and thus likely respond to different drivers. Therefore, we prefer to keep our focus in the manuscript on DC and IR. Notably, as suggested by Reviewer #2, we have added a figure showing the records of CBT' and isoGDGT-related proxies (i.e., BIT and $R_{i/b}$) to the Supplementary materials.*

$$CBT' = {}^{10}log[(Ic + IIa' + IIb' + IIc' + IIIa' + IIIb' + IIIc')/(Ia + IIa + IIIa)] \qquad (1)$$

-The reply letter states that GDGTs were not detected in wind-transported dust. Although this was shown in Hopman's 2004, a later publication (Fietz et al., 2013, doi: 10.1016/j.orggeochem.2013.09.009) shows that GDGTs can be transported by dust. So this part of the manuscript needs to be revised with taking dust-transported GDGTs into account.

*Reply: We thank the reviewer for their comments and providing relevant references. We have added this reference to the manuscript (Line 105-106). Although this reference shows that aeolian transport of brGDGTs is indeed possible, it should be noted that the concentration of brGDGTs in the source region of the dust that accumulates on the Chinese Loess Plateau is below the detection limit, as reported by Gao et al. (2012). Therefore, brGDGT signals in loess-paleosol sequence can be interpreted to reflect the local climate conditions.*

-Add GDGT concentrations to the main manuscript as shown in reply letter fig. 2

*Reply: We thank Dr. Naafs for his comment. We have added the concentration of iso- and brGDGTs to Fig. 2 of the manuscript and included the description in the Results. Line 186-187.*

-As the LR04 is used for tuning, there is some circlular reasoning here. The LR04 stack is tuned to astronomical cycles, so when you tune your record to the LR04, you will be default get astronomical cycles. This caveat needs to be acknowledged. I dont say that this will explain all variance, but it will contribute.

*Reply: We thank Dr. Naafs for his comment. Tuning to LR04 can indeed lead to circular reasoning. However, it is important to note that for our age model, only the loess/paleosol boundaries are tied to glacial/interglacial transitions in the LR04 record. The rest of our age model primarily relies on millennial-scale events that are aligned with speleothem records, which benefit from absolute dating methods (Cheng et al., 2016), and is further independently supported by OSL dates from a nearby loess/paleosol section (see Fuchs et al., 2023 for details).*

*Regarding the spectral analysis results and our discussion on different drivers of loess proxies (100-kyr cycle) and the degree of cyclization (23-kyr cycle): these different proxy records are derived from the same core and analyzed on the same material. In the theoretical case that the 100 kyr cycle is an artifact of the link to the LR04 record, the shorter cycles are truly present. Besides, the presence of astronomical cycles in loess proxy records is not the novelty of our study, but has long been recognized in loess/paleosol sections that were independently dated using paleomagnetism, magnetic susceptibility, grain size, and OSL (e.g., Heller and Tung-Sheng, 1982; Kukla et al., 1988; Stevens et al., 2018). Nevertheless, we have added a line on the possible introduction of a 100 kyr cycle in our records by using this approach. See Line 120-123 in the revised manuscript.*

-I like the addition of fig. 5 from the reply, but what is not clear to me is what "small spectral density" means in the reply. It looks like the density of the IR record is significant for 23-kyr and around 30% of the density of the MS record. This needs to be discussed in the revised manuscript.

*Reply: We agree that IR shows 23-kyr cycle in the spectral analysis. However, we have decided not to elaborate on this information in the revised manuscript, as this record is also part of the discussion and interpretation of the MBT'$_{5Me}$ record in a temperature-focused manuscript that is currently under review with Organic Geochemistry. More importantly, the climatic meaning of IR in this downcore remains unclear, and thus discussing astronomical cycles related to this proxy would not be pertinent at this stage.*

**Referee comment #2:**

The authors responded in an adequate way to the comments of three reviewers. I would only have a last suggestion left: while the decision is explained not to include qualitative proxies for precipitation (Ri/b and BIT index), I think that mentioning in the manuscript text that there is no response of these proxies on the timescale and environmental change-scale adds to our understanding of precipitation proxies in loess in general. The figure presented in the rebuttal could for instance be included as a supplementary figure.

*Reply: We thank the reviewer for their positive evaluation of our revision. We agree that the addition of these proxies ($R_{i/b}$ and BIT) can be informative, and have added these panels to Fig. S3 in the supplementary information.*

**Referee comment #3:**

Guo et al. have substantially improved their manuscript using the comments by David Naafs, Anonymous Referee #2, and myself. The authors' descriptions of applied changes accurately reflect those applied to the revised manuscript. To keep their manuscript to-the-point and focused on quantitative reconstruction of monsoon precipitation with brGDGTs, the authors maintained the focus on the degree of cyclization (DC) and isomer ratio (IR).

Overall, I am happy with the authors' responses and revisions. In my opinion, the authors provided valid reasons for keeping the focus on DC and IR. I thus see no reason to insist on the addition of isoGDGT-based proxies (e.g., BIT index and Ri/b) and other brGDGT-based proxies (e.g., CBT(′) and the precipitation index, PI). Instead, I only have a few, mostly editorial suggestions for further revision, and I would not need to read the manuscript again before publication in Climate of the Past: all line numbers refer to the clean version of the revised manuscript.

*Reply: We thank the reviewer for their positive evaluation of our work and their careful check in the details. Please find our point-by-point response below in italic.*

**Detailed comments:**

Main text

Line 51: Here, Fig. 3 is now first cited before Fig. 1.

*Reply: We have changed this in the revised manuscript.*

Line 259 [Eq. (3)]: As I stated in my previous report, the r value—the coefficient of correlation—is to be reported when describing correlations and I thank the authors for having followed my suggestion. However, when describing regression model results as done here, it is the R2 value—the coefficient of determination which represents the percentage of explained variance—which is to be reported rather than the r value. The authors may also consider reporting the R2 rather than r value in Fig. 4F—but not the other Fig. 4 panels.

*Reply: We thank the reviewer for their detailed explanation of the difference between r value (the coefficient of correlation) and $r^2$ value (the coefficient of determination). We have changed and reported the $r^2$ value for the coefficient of determination in Fig. 4F, Eq. 3 and main text in the revised manuscript. Line 267 and 269.*

Lines 340–341: Should be "(e.g., Clemens et al., 2010)", not "(e.g., (Clemens et al., 2010)Clemens et al., 2010)".

*Reply*: *We thank the reviewer for their detailed check. We have corrected this in the revised manuscript. Line 346.*

Line 360–361 (Data availability): I thank the authors for providing the PANGAEA and Open Science Frame links to their individual GDGT data—individual brGDGT and isoGDGT relative abundances—in response to David Naafs' comment. However, it would have been even better if the authors also provided individual GDGT peak areas and/or concentrations in µg g soil–1, unless the authors prefer to do so when their other manuscript under review with Organic Geochemistry is accepted or published.

*Reply*: *We thank the reviewer for their suggestions. We have added the concentration of GDGTs into the supplementary file archived in the Open Science Frame. But note that the latest round-robin test has claimed that the comparison of concentrations of brGDGTs between labs still remains challenging, although quantification within laboratories was generally consistent (De Jonge et al., 2024).*

Line 525: Should be "Sinninghe Damsté, J. S.", not "Damsté, J. S. S.".

*Reply*: *We have corrected this in the revised manuscript. Line 529.*

**References**

Cheng, H., Edwards, R. L., Sinha, A., Spötl, C., Yi, L., Chen, S., Kelly, M., Kathayat, G., Wang, X., Li, X., Kong, X., Wang, Y., Ning, Y., and Zhang, H.: The Asian monsoon over the past 640,000 years and ice age terminations, Nature, 534, 640–646, https://doi.org/10.1038/nature18591, 2016.

Fuchs, L., Guo, J., Schefuß, E., Sun, Y., Guo, F., Ziegler, M., and Peterse, F.: Isotopic and magnetic proxies are good indicators of millennial-scale variability of the East Asian monsoon, Commun. Earth Environ., 4, 425, https://doi.org/10.1038/s43247-023-01090-z, 2023.

Gao, L., Nie, J., Clemens, S., Liu, W., Sun, J., Zech, R., and Huang, Y.: The importance of solar insolation on the temperature variations for the past 110kyr on the Chinese Loess Plateau, Palaeogeogr. Palaeoclimatol. Palaeoecol., 317–318, 128–133, https://doi.org/10.1016/j.palaeo.2011.12.021, 2012.

Heller, F. and Tung-Sheng, L.: Magnetostratigraphical dating of loess deposits in China, Nature, 300, 431–433, https://doi.org/10.1038/300431a0, 1982.

De Jonge, C., Peterse, F., Nierop, K. G. J., Blattmann, T. M., Alexandre, M., Ansanay-Alex, S., Austin, T., Babin, M., Bard, E., Bauersachs, T., Blewett, J., Boehman, B., Castañeda, I. S., Chen, J., Conti, M. L. G., Contreras, S., Cordes, J., Davtian, N., van Dongen, B., Duncan, B., Elling, F. J., Galy, V., Gao, S., Hefter, J., Hinrichs, K.-U., Helling, M. R., Hoorweg, M., Hopmans, E., Hou, J., Huang, Y., Huguet, A., Jia, G., Karger, C., Keely, B. J., Kusch, S., Li, H., Liang, J., Lipp, J. S., Liu, W., Lu, H., Mangelsdorf, K., Manners, H., Martinez Garcia, A., Menot, G., Mollenhauer, G., Naafs, B. D. A., Naeher, S., O'Connor, L. K., Pearce, E. M., Pearson, A., Rao, Z., Rodrigo-Gámiz, M., Rosendahl, C., Rostek, F., Bao, R., Sanyal, P., Schubotz, F., Scott, W., Sen, R., Sluijs, A., Smittenberg, R., Stefanescu, I., Sun, J., Sutton, P., Tierney, J., Tejos, E., Villanueva, J., Wang, H., Werne, J., Yamamoto, M., Yang, H., and Zhou, A.: Interlaboratory Comparison of Branched GDGT Temperature and pH Proxies Using Soils and Lipid Extracts, Geochemistry, Geophys. Geosystems, 25, 1–17, https://doi.org/10.1029/2024GC011583, 2024.

Kukla, G., Heller, F., Liu Xiu Ming, Xu Tong Chun, Liu Tung Sheng, and An Zhi Sheng:

Pleistocene climates in China dated by magnetic susceptibility, Geology, 16, 811–814, https://doi.org/10.1130/0091-7613(1988)016<0811:PCICDB>2.3.CO;2, 1988.

Stevens, T., Buylaert, J. P., Thiel, C., Újvári, G., Yi, S., Murray, A. S., Frechen, M., and Lu, H.: Ice-volume-forced erosion of the Chinese Loess Plateau global Quaternary stratotype site, Nat. Commun., 9, https://doi.org/10.1038/s41467-018-03329-2, 2018.